# Fourier PINNs: From Strong Boundary Conditions to Adaptive Fourier Bases

**Madison Cooley**                                                               *mcooley@sci.utah.edu*
*Scientific Computing and Imaging Institute*
*University of Utah*

**Varun Shankar**                                                                *shankar@cs.utah.edu*
*Scientific Computing and Imaging Institute*
*University of Utah*

**Robert M. Kirby**                                                              *kirby@cs.utah.edu*
*Scientific Computing and Imaging Institute*
*University of Utah*

**Shandian Zhe**                                                                 *zhe@cs.utah.edu*
*Kahlert School of Computing*
*University of Utah*

**Reviewed on OpenReview:** *https://openreview.net/forum?id=KqRnsEMYLx*

## Abstract

Interest is rising in Physics-Informed Neural Networks (PINNs) as a mesh-free alternative to traditional numerical solvers for partial differential equations (PDEs). However, PINNs often struggle to learn high-frequency and multi-scale target solutions. To tackle this problem, we first study a strong Boundary Condition (BC) version of PINNs for Dirichlet BCs and observe a consistent decline in relative error compared to the standard PINNs. We then perform a theoretical analysis based on the Fourier transform and convolution theorem. We find that strong BC PINNs can better learn the amplitudes of high-frequency components of the target solutions. However, constructing the architecture for strong BC PINNs is difficult for many BCs and domain geometries. Enlightened by our theoretical analysis, we propose Fourier PINNs — a simple, general, yet powerful method that augments PINNs with pre-specified, dense Fourier bases. Our proposed architecture likewise learns high-frequency components better but places no restrictions on the particular BCs or problem domains. We develop an adaptive learning and basis selection algorithm via alternating neural net basis optimization, Fourier and neural net basis coefficient estimation, and coefficient truncation. This scheme can flexibly identify the significant frequencies while weakening the nominal frequencies to better capture the target solution's power spectrum. We show the advantage of our approach through a set of systematic experiments.

## 1 Introduction

Physics-informed neural networks (PINNs) (Raissi et al., 2019) are innovative, mesh-free approaches for solving partial differential equations (PDEs). They offer alternatives to traditional mesh-based numerical methods such as finite elements and finite volumes (Reddy, 2019). The optimization of PINNs involves *softly* constraining neural networks (NNs) through customized loss functions designed to adhere to the governing equations of physical processes. Researchers have successfully applied PINNs in various domains— for example, they have been used to simulate the radiative transport equation (Mishra & Molinaro, 2021), which is crucial for radio frequency chip and material design (Chen et al., 2020; Liu & Wang, 2019). Further,

cardiovascular flow modeling (Kissas et al., 2020) is another application, along with various fluid mechanics problems (Cai et al., 2021), and high-speed aerodynamic flow modeling (Mao et al., 2020), among many others (Raissi et al., 2020; Chen et al., 2020; Jin et al., 2021; Sirignano & Spiliopoulos, 2018; Zhu et al., 2019; Geneva & Zabaras, 2020; Sahli Costabal et al., 2020; Sun et al., 2020; Fang & Zhan, 2019).

Despite these successes, the training of PINNs remains challenging in some instances. Recent studies have analyzed common failure modes of PINNs, particularly when modeling problems that exhibit high-frequency, multi-scale, chaotic, or turbulent behaviors (Wang et al., 2022; 2021c;b; 2024b), or when the governing PDEs are stiff (Krishnapriyan et al., 2021; Mojgani et al., 2023). Rahaman et al. (2019) attributes the slow convergence to the high-frequency components of the target solution, identifying it as a "spectrum bias" in standard NNs, while learning low-frequency information from data is straightforward. Wang et al. (2021c) confirmed that this bias is also present in the PINN setting. These challenges often arise because applying differential operators over the NN in the residual complicates the loss landscape (Krishnapriyan et al., 2021). From an optimization perspective, Wang et al. (2021b) highlighted that the imbalance in gradient magnitudes between the boundary loss and residual loss (with the latter often being much larger) causes the residual loss to dominate training, leading to a poor fit to the boundary conditions. Wang et al. (2022) confirmed this conclusion through a neural tangent kernel (NTK) analysis of PINNs with wide networks, finding that the dominant eigenvalues of the residual kernel matrix often result in the training primarily fitting the residual loss.

One class of approaches designed to mitigate the training challenges in PINNs involves setting different weights for the boundary and residual loss terms. For example, Wight & Zhao (2020) suggested incorporating a large multiplier for the boundary loss term to prevent the residual loss from dominating the training process. In contrast, Wang et al. (2021b) proposed a dynamic weighting scheme based on the gradient statistics of the loss terms, and Wang et al. (2022) developed an adaptive weighting approach based on the eigenvalues of the NTK. Liu & Wang (2021) employed a mini-max optimization to update the loss weights via stochastic ascent, and McClenny & Braga-Neto (2024) used a multiplicative soft attention mask to dynamically re-weight the loss term for each data point and collocation point. To alleviate the spectrum bias in NNs, Tancik et al. (2020) randomly sampled a set of high frequencies from a large-variance Gaussian distribution to construct random Fourier features as input to the NN. Additionally, Wang et al. (2021c) used multiple Gaussian variances to sample frequencies for Fourier features, aiming to capture multi-scale solution information within the PINN framework. While effective, the performance of this method is sensitive to the number and scales of the Gaussian variances, which are user-specified hyperparameters that are often difficult to optimize.

Another strategy to improve these challenges is to modify the NN architecture to exactly satisfy the boundary conditions (BCs) (Lu et al., 2021; Lyu et al., 2021; Lagaris et al., 1998; 2000; McFall & Mahan, 2009; Berg & Nyström, 2018; Lagari et al., 2020; Mojgani et al., 2023). We refer to these approaches as "strong BC PINNs". Despite their effectiveness, these approaches face several limitations. They are usually limited to tasks with relatively simple and well-defined physics, requiring significant craftsmanship and complex implementations even in straightforward problem settings. Consequently, these strategies are less flexible than the original PINN framework, which employs a soft constraint approach for boundary condition satisfaction. Additional complications arise for challenging physical systems governed by invariances or conservation laws, such as energy or momentum conservation, due to often poorly understood and imprecisely defined physical laws. Incorporating these laws effectively into the NN architecture makes extending strong BC PINNs to handle more complex tasks difficult. Further, designing a custom ansatz requires tailoring the NN architecture for each specific boundary condition and domain, which is often impractical for complicated domains. However, when properly designed, these techniques can achieve highly accurate solutions.

In our work, we delve deeper into the training challenges of PINNs for learning high-frequency and multi-scale solutions. We analyze the mechanisms behind the success of strong BC PINNs and explore ways to incorporate these successful strategies into a more general PINN architecture. Our specific contributions are as follows:

- We first examine a strong BC PINN architecture for simple Dirichlet boundary conditions as proposed by Lu et al. (2021). This variant integrates a fixed polynomial boundary function into the NN architecture to exactly satisfy the boundary conditions. While it shows significant improvements

over the standard PINN, especially for higher frequency problems, it struggles to predict solutions with frequencies above a certain threshold due to its static nature. To address this limitation, we propose a new strong BC PINN architecture featuring an adaptive parameter optimized during training. This parameter adjusts the boundary function's sharpness to match the true solution, thereby improving accuracy for higher-frequency solutions compared to the static polynomial variant.

- We conduct a Fourier analysis on both strong BC PINNs compared to the standard PINN. Through the Fourier series convolution theory, we discovered that multiplying the NN by the strong boundary function significantly enhances the learning speed and accuracy of the higher frequency coefficients in the target solution. In contrast, standard PINNs struggle to accurately capture coefficients in the high-frequency domain. This analysis complements and confirms the aforementioned NTK work.

- Inspired by our Fourier analysis, we develop Fourier PINNs. This novel PINN architecture enhances frequency learning within the true solution, comparable to strong BC PINNs, regardless of specific boundary conditions, domain, or underlying physical properties. The Fourier PINN architecture integrates a standard NN with a linear combination of Fourier bases, with frequencies uniformly sampled from an extensive pre-set range. We implement an adaptive learning and basis selection algorithm that alternately optimizes the NN basis parameters and the coefficients of the NN and Fourier bases while pruning insignificant bases. This approach efficiently identifies significant frequencies, supplements those missed by the NN, and improves frequency amplitude estimation (i.e., the basis coefficient) while maintaining computational efficiency. Unlike previous methods, this approach only requires specifying a sufficiently large range and small spacing for the Fourier bases without concern for the actual number and scales of frequencies in the true solution.

- We evaluate Fourier PINNs on several benchmark PDEs characterized by high-frequency and multi-frequency solutions. In all cases, Fourier PINNs consistently achieve low solution errors (e.g., $\sim 10^{-3}$ or $\sim 10^{-4}$). In contrast, standard PINNs invariably fail to achieve comparable accuracy. PINNs with random Fourier features (RFF-PINNs) often fail across various Gaussian variances and scales, indicating high sensitivity to these parameters. Additionally, we test spectral methods, PINNs with large boundary loss weights (Wight & Zhao, 2020), and PINNs with adaptive activation functions (Jagtap et al., 2020). Fourier PINNs consistently outperform all these methods.

The remainder of this paper is structured as follows: Section 2 provides the necessary background and notation, while in Section 4, we present our Fourier analysis of strong BC PINNs and explain the success of the strong boundary ansatz methodology. Our new Fourier PINN architecture and its training routines are described in Section 5. Section 6 details our numerical experiments and findings, including assessments of computational cost and accuracy compared to baseline methods. Finally, Section 7 discusses future work regarding scalability and Section 8 discusses the results and outlines specific future research directions.

## 2 Background

In this section, we first describe the general optimization problems we are addressing with physics-informed neural networks (PINNs) following the formulation presented in Raissi et al. (2019). We then discuss the specifics of strong boundary condition enforcement.

### 2.1 General Overview of Physics-Informed Neural Networks

The PINN framework uses NNs to estimate solutions to partial differential equations (PDEs). Consider a PDE of the following general form,

$$\mathcal{F}[u](\mathbf{x}) = f(\mathbf{x}), \;\; \mathbf{x} \in \Omega, \tag{1}$$

where $\mathcal{F}$ is a linear or non-linear differential operator and $u$ represents the unknown solution and $\Omega$ is the problem domain in $\mathbb{R}^d$. The general boundary conditions are then,

$$\mathcal{B}[u](\mathbf{x}) = g(\mathbf{x}), \;\; \mathbf{x} \in \partial\Omega, \tag{2}$$

where $\partial\Omega$ is the boundary of the domain and $\mathcal{B}$ is a general boundary-condition operator.

To solve the PDE, the PINN uses a deep NN, $u_N(\mathbf{x};\boldsymbol{\theta})$, to approximate the true solution $u(\mathbf{x})$. For clarity in later sections, we follow the convention of Cyr et al. (2020) and define the output of the NN with width $W$ as a linear combination of a set of nonlinear bases such that

$$u_N(\mathbf{x};\mathbf{c},\boldsymbol{\theta}^H) = \sum_{j=1}^{W} c_j \boldsymbol{\psi}_j(\mathbf{x};\boldsymbol{\theta}^H), \tag{3}$$

where each $\boldsymbol{\psi}_j$ are nonlinear activation functions (such as Tanh) acting on the hidden layer outputs. Each $c_j$ for $j = 1, ..., w$ and $\boldsymbol{\theta}^H$ are the weights and biases in the last layer of the NN and hidden layers, respectively. Therefore, $\boldsymbol{\theta} = \{\mathbf{c}, \boldsymbol{\theta}^H\}$ form the set of all network parameters. Then, finding the optimal network parameters $\boldsymbol{\theta}^*$ involves minimizing the following composite loss function between boundary and residual loss terms,

$$\boldsymbol{\theta}^* = \arg\min_{\boldsymbol{\theta}} \quad \lambda L_b(\boldsymbol{\theta}) + L_r(\boldsymbol{\theta}). \tag{4}$$

Here,

$$L_b(\boldsymbol{\theta}) = \frac{1}{M} \sum_{i=1}^{M} \left( \mathcal{B}[u_N](\mathbf{x}_b^i) - g(\mathbf{x}_b^i) \right)^2, \tag{5}$$

is the boundary loss to fit the boundary condition with Lagrange multiplier $\lambda$, and

$$L_r(\boldsymbol{\theta}) = \frac{1}{N} \sum_{i=1}^{N} \left( \mathcal{F}[u_N](\mathbf{x}_r^i) - f(\mathbf{x}_r^i) \right)^2, \tag{6}$$

is the residual loss to fit the equation. We minimize Equation 4 by sampling $N$ collocation points $\{\mathbf{x}_r^i\}_{i=1}^{N}$ from the domain $\Omega$ and $M$ points $\{\mathbf{x}_b^i\}_{i=1}^{M}$ from the boundary of the domain $\partial\Omega$, and iteratively modifying the network parameters through gradient descent.

## 2.2 Strong Boundary Condition PINNs

Equation 5 can approximately enforce various types of boundary conditions, including Dirichlet, Neumann, Robin, and periodic. However, prior analysis by Wang et al. (2021b; 2022) suggests that the instability in training PINNs likely arises from the competition between the weakly enforced boundary loss (5) and the residual loss (6) terms during optimization. This competition likely occurs because both loss terms are minimized simultaneously during training. However, the optimization algorithm tends to prioritize the minimization of the residual loss, as it typically dominates the gradient of the combined loss function. Consequently, this can lead to a scenario where the residual loss converges effectively but at the expense of the boundary loss, which remains inadequately optimized. As a result, the boundary conditions may not be satisfied, leading to poor model performance on new, unseen data or data at the domain's boundaries.

One solution to mitigate the competition between the loss terms is to design a surrogate model that inherently satisfies the boundary conditions. This approach typically works by modifying the network architecture to satisfy the boundary conditions exactly, thus eliminating the need for a boundary enforcement loss term and further reducing the computational cost. This method of exact imposition of boundary conditions has become a standard approach in the PINN literature and is especially prevalent for Dirichlet and periodic boundary conditions (Lu et al., 2021; Yu et al., 2022; Wang et al., 2024a).

This work primarily focuses on Dirichlet boundary conditions to simplify the analysis and implementations while addressing a significant and common scenario in physical systems. Dirichlet boundary conditions provide a clear and straightforward framework to demonstrate the effectiveness of the surrogate model approach, as they require the solution to take specific values at the boundaries, which is relatively easy to enforce exactly on simplified domains. To illustrate, we consider the 1D Dirichlet boundary condition defined as,

$$x \in [a, b], \quad u(a) = u(b) = g(x). \tag{7}$$

For this case, the surrogate model is typically defined as,

$$u_{\boldsymbol{\theta}}(x) = g(x) + \phi(x)u_N(x), \tag{8}$$

such that $u_{\boldsymbol{\theta}}(x)$ is the solution function, $g(x)$ is the provided boundary function, $u_N(x)$ is the output of the neural network, and $\phi(x)$ is a composite distance function that zeroes out on the boundaries, ensuring the boundary conditions are met without explicit penalty terms. By construction, $u_{\boldsymbol{\theta}}(a) = u_{\boldsymbol{\theta}}(b) = g(x)$, and thus the boundary condition in Equation 7 is strictly and automatically satisfied. To estimate the parameters of $u_{\boldsymbol{\theta}}$, we only need to minimize the residual loss $L_r(\boldsymbol{\theta})$. Most PINN surrogate model designs rely on distance functions based on the theory of R-functions (Sukumar & Srivastava, 2022), which are smooth mathematical functions that encode Boolean logic and facilitate the combination of simple shapes to form complex geometries (Rvachev & Sheiko, 1995). Some meshfree Galerkin methods have utilized the capability of R-functions to enforce boundary conditions smoothly and exactly to enhance the precision and robustness of numerical simulations for solving boundary-value problems (Shapiro & Tsukanov, 1999; Akin, 1994; Tsukanov & Posireddy, 2011).

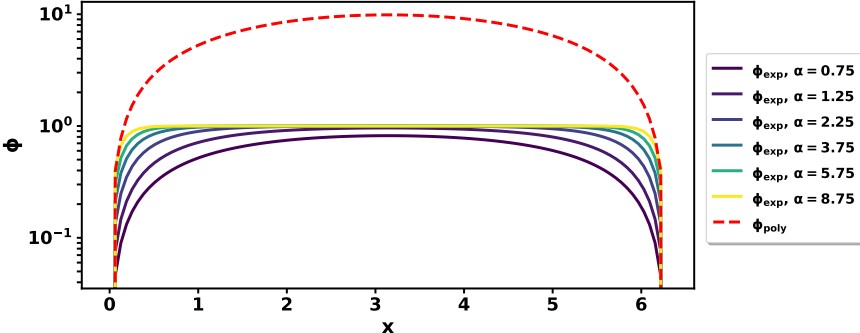

Figure 1: *This figure shows the polynomial distance function $\phi_{poly}(x)$ and the exponential distance function $\phi_{exp}(x)$ with different $\alpha$ values. The curves for $\phi_{exp}(x)$ correspond to different $\alpha$ parameters, demonstrating how the shape of $\phi_{exp}(x)$ "sharpens" around the domain boundaries as $\alpha$ increases.*

In this work, we compare the standard PINN to two strong BC PINNs using different distance functions to enforce Dirichlet boundary conditions exactly. The first distance function we consider is the polynomial distance function proposed by (Lu et al., 2021), defined as,

$$\phi_{poly}(x) = (x - a)(b - x). \tag{9}$$

This function satisfies the Dirichlet boundary conditions by zeroing out at the $x = a$ and $x = b$ endpoints. While Lu et al. (2021) demonstrated promising results using this formulation, the polynomial distance function $\phi_{poly}(x)$ remains static throughout the training process. The lack of adaptability limits the model's ability to handle different problem domains and boundary conditions, which can hinder performance when faced with varying complexities within the solution space.

To address this limitation, we propose an adaptive distance function that introduces flexibility in enforcing boundary conditions, which we define as:

$$\phi_{exp}(x) = (1 - e^{\alpha_a(a-x)})(1 - e^{\alpha_b(x-b)}), \tag{10}$$

where $\alpha_a$ and $\alpha_b$ are parameters that can be pre-set or optimized during training. These parameters allow the function to adjust to the characteristics of the problem domain dynamically, ensuring robust enforcement of boundary conditions across different scenarios. As we demonstrate in subsequent sections, this flexibility improves accuracy and enhances the efficiency of the training process by enabling the network to adapt to the complexity of the solution space. Allowing independent values for $\alpha_a$ and $\alpha_b$ enables additional flexibility and allows the function to adapt asymmetrically to different boundary conditions. Although the proposed adaptive distance function in Equation 10 may appear incremental, its primary contribution lies in demonstrating a scalable and flexible framework for enforcing strong boundary conditions. This method

significantly improves over traditional PINNs and non-adaptive distance functions, particularly for problems that require the exact imposition of Dirichlet boundary conditions. Additionally, in later sections, we present a thorough theoretical analysis that elucidates the mechanisms driving the improvements in enforcing strong boundary conditions, further highlighting the advantages of the proposed method.

The general form of the adaptive distance function for a $d$-dimensional rectilinear domain $\Omega \subset \mathbb{R}^d$, with spatial coordinates $\mathbf{x} = (x_1, x_2, ..., x_d)$, is:

$$\phi_{\exp}(\mathbf{x}) = \prod_{i=1}^{d} \left(1 - e^{\alpha_{a_i}(a_i - x_i)}\right) \left(1 - e^{\alpha_{b_i}(x_i - b_i)}\right), \tag{11}$$

where $a_i$, $b_i$ represent the boundaries in the $i$-th dimension, and $\alpha_{a_i}, \alpha_{b_i}$ are the corresponding steepness parameters. This form ensures that the boundary conditions are enforced independently in each dimension, making the approach easily scalable to higher dimensions in rectilinear domains.

One potential strategy for adaptation for irregular domains is using Fourier extension methods, which allow non-periodic functions defined on arbitrary domains to be extended into periodic ones. Doing so can enforce the boundary conditions in a more structured domain, allowing the network to handle irregularities better. Another promising direction is leveraging custom network layers that map irregular domains to rectilinear spaces, as demonstrated by methods such as the GOFNO network Liu et al. (2023). Extending this framework to more complex boundary conditions, such as Neumann or Robin conditions, is an important future direction. The adaptive distance function can be modified for Neumann boundary conditions to have non-zero slopes at the boundaries, ensuring correctly enforced derivatives. This would require careful and non-trivial adjustments to the distance function, but the general framework remains applicable.

Figure 1 illustrates the behavior of the polynomial distance function $\phi_{poly}(x)$ to the exponential distance function $\phi_{exp}(x)$ with various values of $\alpha$. As $\alpha$ varies, $\phi_{exp}(x)$ adjusts its shape, showcasing its ability to conform to different problem domains and complexities. We expect this dynamic adaptability to enhance the model's performance by more precisely enforcing boundary conditions across diverse scenarios. We affirm this through a numerical example in the next section.

## 3 Boundary Condition Pathologies in PINNs

Recent works have employed strong boundary enforcement strategies (Lu et al., 2021; Yu et al., 2022) but offer few insights into the specific surrogate model design process. The design and implementation of the surrogate models are often ad-hoc, with no systematic approach to their architectural design. This complicates their application and limits their effectiveness across boundary conditions and physical problems, especially when dealing with complex physics and domains. Moreover, the standard PINNs frequently encounter difficulties with high-frequency and multi-scale solutions, leading to inaccurate predictions (Wang et al., 2021c). We show that while the strong BC PINN model improves solution quality for problems with higher frequency compared to the standard PINN, the solution quality nevertheless degrades with larger frequencies. In the following, we empirically demonstrate how the standard PINN solution quality degrades as the actual frequency of the solution increases in both a linear 1D Poisson problem and a non-linear 1D steady-state Allen-Cahn problem. Specifically, we consider the fabricated solution $u(x) = \sin(kx)$ for a range of frequencies $k \in [2, 6, 10, 14, 18, 22, 26, 30, 34]$ and $x \in [0, 2\pi]$. Therefore, by varying $k$, we can examine the performance of the standard PINNs compared to the strong BC PINNs when the solution $u$ includes different frequency information.

Specifically, we examine the 1D Poisson equation given by,

$$\Delta u = f(x) \tag{12}$$

$$u(0) = u(1) = 0, \tag{13}$$

and derive the forcing function $f(x)$ from the fabricated solution $u(x) = \sin(kx)$ which gives $f(x) = -k^2 \sin(kx)$. We also consider the 1D Allen-Cahn equation defined as,

$$u_{xx} + u(u^2 - 1) = f(x) \tag{14}$$
$$u(0) = u(1) = 0, \tag{15}$$

which gives the forcing function $f(x) = -k^2 \sin(kx) + \sin(kx)(\sin^2(kx) - 1)$. Both are subject to the Dirichlet condition in Equation 7 where $a = 0$ and $b = 2\pi$.

We test these problems using a standard PINN, which explicitly incorporates the weakly enforced boundary loss and residual loss terms. We compare these results to two strong BC PINNs employing different distance functions (8) and show that this method exhibits slower solution degradation with increased frequency. Specifically, we tested two neural network architectures, each with 100 neurons per layer and depths of two and four, and assessed model performance using the relative $\ell_2$ error, defined as

$$\frac{\|y - \hat{y}\|_2}{\|y\|_2} = \frac{\sqrt{\sum_{i=1}^{n}(y_i - \hat{y}_i)^2}}{\sqrt{\sum_{i=1}^{n} y_i^2}},$$

where $\hat{y}$ is the predicted solution and $y$ is the true solution. All networks used the `Tanh` activation and are trained using 10K equally spaced collocation points sampled from the domain. Relative $\ell_2$ errors are reported on a separate testing set of 20K points. All experiments used 32-bit floating-point precision (float32) to ensure computational efficiency. Each model was trained for 100K iterations using the Adam optimizer (Kingma & Ba, 2015) with an initial learning rate of $10^{-3}$, decaying exponentially by 0.9 every 1000 iterations, followed by L-BFGS optimization until convergence (tolerance $10^{-9}$). For the strong BC PINN with $\phi_{\exp}$, we initialized $\alpha$ to 0.5 and jointly optimized it with all other parameters. Results are averaged over five random trials and were conducted on a GeForce RTX 3090 GPU with CUDA version 12.3, running on Ubuntu 20.04.6 LTS. Additionally, we implemented a second-order accurate Finite Difference Method (FDM) as a baseline for solving the 1D Poisson and steady-state Allen-Cahn equations using 1000 discretization points and central differences. Table 3 in the Appendix provides detailed hyperparameters and experimental design information.

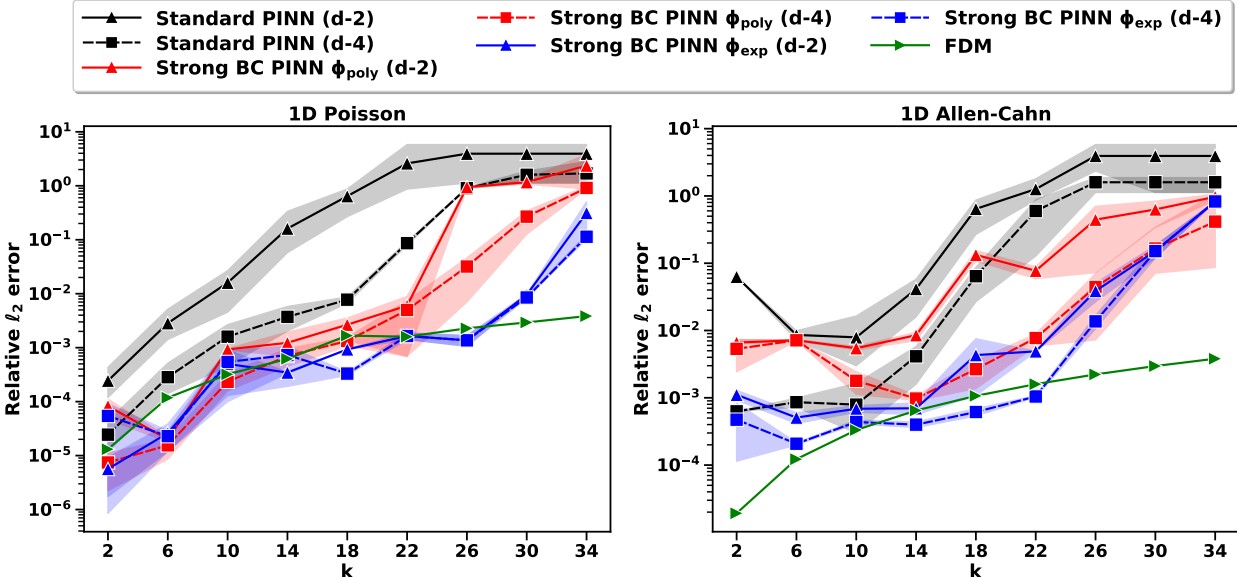

Figure 2: *Relative $\ell_2$ error in predicting the solution $u(x) = \sin(kx)$ as a function of frequency $k$. The plots compare the performance of the standard PINN, the strong BC PINN with polynomial boundary function $\phi_{poly}$, the strong BC PINN with exponential boundary function $\phi_{exp}$, and FDM for (left) 1D Poisson and (right) 1D steady-state Allen-Cahn equations for NN with 2 hidden layers (d-2) and 4 hidden layers (d-4) each with 100 neurons. The shaded regions represent the error variability.*

Figure 2 demonstrates that both strong BC PINNs, using $\phi_{\text{poly}}$ and $\phi_{\text{exp}}$ boundary functions, effectively suppress high-frequency components, leading to higher accuracy solutions compared to the standard PINN. The deeper architecture (d=4) consistently reduces errors across most frequencies for both boundary functions, but the static polynomial distance function $\phi_{\text{poly}}$ shows diminishing benefits as the frequency increases. In contrast, the strong BC PINN with $\phi_{\text{exp}}$ maintains superior accuracy, outperforming both the standard PINN and the strong BC PINN with $\phi_{\text{poly}}$ across a broader frequency range. Although the advantages of strong boundary enforcement diminish at very high frequencies, $\phi_{\text{exp}}$ consistently retains its edge over $\phi_{\text{poly}}$.

Additionally, the Finite Difference Method (FDM) achieves the lowest relative $\ell_2$ errors across all frequencies in both the 1D Poisson and 1D Allen-Cahn problems, serving as a benchmark for evaluating neural network approaches. The FDM demonstrates exceptional performance, particularly at high frequencies, highlighting the limitations of PINN-based approaches in capturing rapid oscillations. However, the strong BC PINNs with $\phi_{\text{exp}}$ narrow the gap with FDM at lower and mid-range frequencies, showcasing their robustness compared to standard PINNs.

While the FDM is a reliable and efficient approach for solving partial differential equations, it has notable limitations that drive the need for advancements in Physics-Informed Neural Networks (PINNs). FDM relies on structured grids, making it challenging to apply to problems with complex geometries, and it struggles with scalability in high-dimensional problems due to exponential grid growth. Unlike FDM, PINNs can also integrate noisy or partial data into their models, making them more flexible for real-world applications. Furthermore, advancements in machine learning architectures and hardware accelerators have significantly improved the practicality and efficiency of PINNs. By addressing current challenges, such as resolving high-frequency solutions and improving convergence, PINNs have the potential to provide a scalable, versatile alternative to traditional numerical methods like FDM.

Table 1: *Training times (in seconds) for various network configurations across 1D Poisson and 1D Allen-Cahn problems. The values in parentheses represent the standard deviations of training times.*

| Network | 1D Poisson (sec) | 1D Allen-Cahn (sec) |
|---|---|---|
| Standard PINN (d=2) | 202.07 (2.52) | 103.11 (0.10) |
| Standard PINN (d=4) | 319.09 (1.61) | 200.14 (91.20) |
| Strong BC PINN $\phi_{\text{poly}}$ (d=2) | 204.28 (1.95) | 213.41 (78.60) |
| Strong BC PINN $\phi_{\text{poly}}$ (d=4) | 311.83 (10.83) | 122.91 (15.90) |
| Strong BC PINN $\phi_{\text{exp}}$ (d=2) | 299.56 (2.18) | 403.60 (260.00) |
| Strong BC PINN $\phi_{\text{exp}}$ (d=4) | 401.63 (1.98) | 472.59 (56.00) |
| FDM | $< 1$ | $< 1$ |

## 4 Fourier Analysis of Strong BC PINNs

Both NNs and PINNs are known to exhibit spectral bias, meaning they readily capture low-frequency components of a target function but struggle with high-frequency details (Basri et al., 2020; Rahaman et al., 2019; Xu et al., 2020; Wang et al., 2021c). As demonstrated earlier, this bias causes the solution quality of PINNs to degrade as the target solution's frequency increases. However, strong BC PINNs appear to mitigate or delay this issue, achieving better performance for higher-frequency solutions.

This section uses Fourier analysis to investigate why strong BC PINNs outperform standard PINNs in addressing spectral bias. The infinite Fourier series expresses a periodic function $u(x)$ with period $L$ as:

$$u_\infty(x) = \sum_{n=-\infty}^{\infty} \widehat{u}[n]e^{i2\pi nx/L}, \tag{16}$$

where the Fourier coefficients are given by:

$$\widehat{u}[n] = \frac{1}{L}\int_0^L u(x)e^{-i2\pi nx/L}dx. \tag{17}$$

## 4.1 Numerical Fourier Analysis of Strong BC PINNs

To analyze spectral bias, we compute the Fourier coefficients for several manufactured solutions: 1. A single sine wave to demonstrate the baseline frequency handling of PINNs. 2. A combination of sine waves with differing frequencies to explore how multi-scale components are captured. 3. A Gaussian-modulated sine wave to evaluate performance on solutions with continuous spectral content. 4. A sine wave and polynomial combination to examine behavior under non-homogeneous boundary conditions.

For standard PINNs, the frequency spectrum typically shows a pronounced low-frequency peak, reflecting effective learning of low-frequency components and a gradual decay for higher frequencies, often resulting in heavy tails. These heavy tails indicate residual noise and the network's difficulty in accurately filtering out high-frequency inaccuracies. By contrast, as we will show, strong BC PINNs exhibit faster decay in the higher-frequency components, enabling better suppression of noise and improved representation of fine details in the solution.

We trained a standard PINN and a strong BC PINN for each example with four hidden layers, 100 nodes per layer, and `Tanh` activations. Table 4 lists the specific training and hyperparameter details. We then computed the Discrete Fourier Transform (DFT) of the learned solutions $u_\theta(x)$, which represents their frequency spectrum. For $N$ equally spaced points $x_j$ sampled over the domain $[0, 2\pi]$, the DFT is defined as:

$$\widehat{u}_\theta[k] = \sum_{n=0}^{N-1} u_\theta(x_n)e^{-i2\pi kn/N}, \quad k = -\frac{N}{2}, \dots, \frac{N}{2} - 1.$$

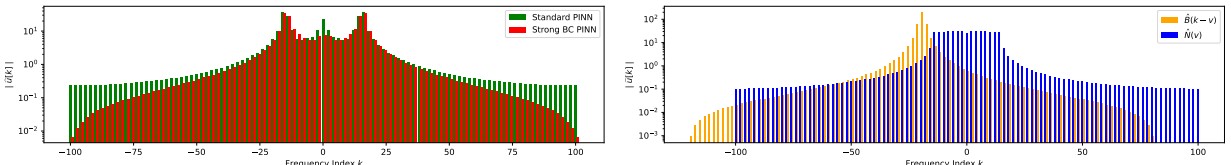

Figure 3: *Frequency spectrum of the learned solution (left) and the convolution operation (right) in the strong BC PINN and standard PINN. The ground-truth solution is $\sin(kx)$ with $k = 15$. The left graph shows frequency handling by standard and strong BC PINNs, while the right demonstrates how convolution reduces high-frequency noise.*

Figure 3 (left) compares the frequency spectrum of a standard PINN and a strong BC PINN trained on the 1D Poisson problem with the ground-truth solution $u(x) = \sin(kx)$, for $x \in [0, 2\pi]$ and $k = 15$. While the standard PINN successfully captures the target frequency, it also retains significant high-frequency noise, as evident in the heavy tails of the spectrum. In contrast, the strong BC PINN exhibits much faster decay in higher frequencies, effectively reducing high-frequency noise and improving accuracy. This indicates that the strong BC PINN filters out unnecessary components more efficiently, enhancing the clarity and precision of the learned solution by isolating the dominant frequency.

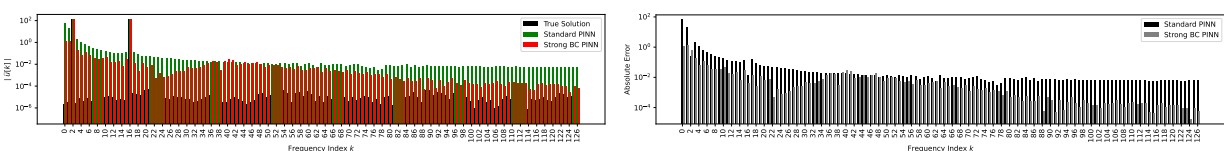

Figure 4: *The frequency spectrum of the learned solutions (top) and absolute error of the Fourier coefficients (bottom) for the strong BC PINN and standard PINN compared to the ground truth. The ground-truth solution is $u(x) = \sin(2x) + \sin(16x)$.*

Building on the analysis of single-frequency solutions in the previous section, we now examine the performance of standard and strong BC PINNs on a multi-scale target solution where multiple frequencies are present simultaneously. This setup provides a more challenging scenario, testing each model's ability to balance between capturing dominant low-frequency components and accurately representing higher-frequency details. For a multi-scale target solution, $u(x) = \sin(2x) + \sin(16x)$, Figure 4 illustrates the frequency spectrum of the

learned solutions. To simplify the analysis and focus on the unique information, we transition to visualizing only the positive frequency components, as the Fourier coefficients are symmetric for real-valued functions.

While the standard PINN captures the primary low-frequency components, it struggles to filter the higher-frequency components effectively, allowing residual high-frequency noise to persist. The strong BC PINN, however, captures both frequencies more accurately and suppresses extraneous high-frequency noise, leading to a lower overall error. This result highlights the strong BC PINN's advantage in multi-scale problems, where it can effectively balance between capturing necessary frequencies and eliminating noise.

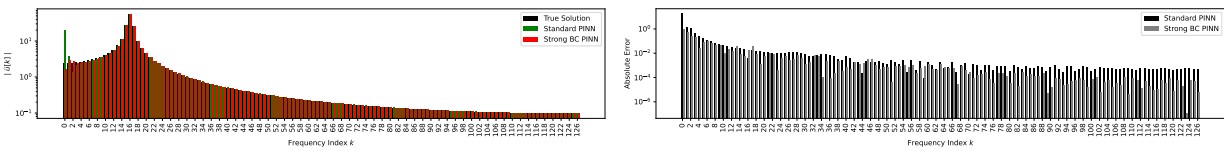

Figure 5: *The frequency spectrum of the learned solutions (top) and absolute error of the Fourier coefficients (bottom) for the strong BC PINN and standard PINN compared to the ground truth. The ground-truth solution is* $u(x) = e^{-0.5x^2} \sin(16x)$.

To evaluate the strong BC PINN's performance on solutions with continuous spectral content, we used a Gaussian-modulated sinusoid, $u(x) = e^{-0.5x^2} \sin(16x)$. Figure 5 shows the Fourier spectrum for this case. The Gaussian modulation introduces a broad spectrum of frequencies, challenging the model's ability to suppress irrelevant high frequencies while accurately capturing the primary frequency content. In this example, the strong BC PINN demonstrates robust noise reduction by filtering unnecessary frequencies, including low frequencies. This capability is essential for handling continuous spectra. The strong BC PINN can generalize well to functions with non-discrete frequency components, making it applicable to a broader range of real-world problems.

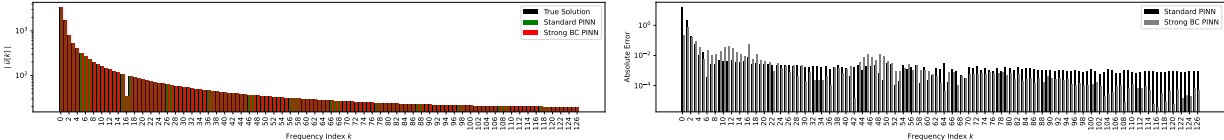

Figure 6: *The frequency spectrum of the learned solutions (top) and absolute error of the Fourier coefficients (bottom) for the strong BC PINN and standard PINN compared to the ground truth. The ground-truth solution is* $u(x) = x^2 + \sin(16x)$.

Finally, we explore the strong BC PINN's performance on non-homogeneous boundary conditions using the manufactured solution $u(x) = x^2 + \sin(16x)$. Figure 6 illustrates the Fourier spectrum for this non-homogeneous boundary case. The polynomial term $x^2$ introduces a low-frequency component that is not harmonically related to the oscillatory term $\sin(16x)$. The standard PINN struggles to capture this low-frequency component fully. At the same time, the strong BC PINN demonstrates improved performance in accurately representing both the polynomial boundary effect and the oscillatory component. This example underscores the strong BC PINN's effectiveness in handling non-homogeneous boundaries, as it can accommodate mixed low- and high-frequency components in the solution.

These results collectively illustrate the strong BC PINN's enhanced ability to manage various frequency cases, from simple single-frequency cases to continuous spectra and complex non-homogeneous boundary conditions. The strong BC PINN demonstrates robustness and flexibility by effectively suppressing high-frequency noise and maintaining accuracy across various spectral components, making it suitable for a wide array of boundary-conditioned problems in applied settings.

## 4.2 Analytical Fourier Analysis of Strong BC PINNs

To analyze why multiplying the distance function ($\phi(x)$) with the NN in Equation 8 helps to obtain more accurate coefficients for high frequencies, we first represent the general distance function $\phi(x)$ as an infinite

Fourier series in its discrete form[1] as in Equation 16,

$$\phi(x) = \phi_\infty(x) = \sum_{n=-\infty}^{+\infty} \hat{\phi}[n] \cdot e^{inx} \tag{18}$$

with period $L = 2\pi$. We first analyze the polynomial distance function by substituting Equation 9 for $\hat{\phi}$ in Equation 18. From Equation 17, we get the Fourier coefficients,

$$\hat{\phi}_{poly}[n] = \frac{1}{2\pi} \int_{2\pi} x(2\pi - x) \cdot e^{-inx} \mathrm{d}x \;=\; -\frac{2}{n^2}. \tag{19}$$

Similarly, to compare against our proposed adaptive distance function, we substitute in Equation 10 for $\hat{\phi}$ in Equation 18 and obtain the Fourier coefficients,

$$\hat{\phi}_{exp}[n] = \frac{1}{2\pi} \int_{2\pi} (1 - e^{-\alpha x})(1 - e^{\alpha(x-2\pi)}) \cdot e^{-inx} \mathrm{d}x \;=\; \frac{\alpha(1 - e^{2\pi\alpha})e^{-2\pi\alpha}}{\pi(\alpha^2 + n^2)}. \tag{20}$$

Here, we expand the boundary functions $\phi(x)$ to be periodic (i.e., duplicating its definition in $[0, 2\pi]$ to other intervals), and the period is $2\pi$. We can now obtain the Fourier transforms of each $\phi_\infty(x)$ by evaluating,

$$\hat{\phi}(\omega) = \sum_{n=-\infty}^{+\infty} \hat{\phi}[n] \cdot \delta\left(\omega - \frac{n}{2\pi}\right) \tag{21}$$

where $\delta(\cdot)$ is the Dirac delta function and $\omega$ is the continuous frequency. Specifically, each $\hat{\phi}(\omega)$ represents the Fourier transform of $\phi_\infty(x)$, meaning $\mathcal{F}^{-1}\left\{\hat{\phi}(\omega)\right\} = \phi_\infty(x)$ where $\mathcal{F}^{-1}$ denotes the inverse Fourier transform.

Since $|\hat{\phi}_{poly}[n]| \propto 1/n^2$, the amplitude of the frequencies of $\phi_{poly}(x)$ decay quadratically fast with the increase of the absolute frequency value. This rapid decay is beneficial for denoising as it effectively suppresses high-frequencies, leading to smoother approximations. However, it may result in the loss of fine details, especially if essential signal components are present at higher frequencies. Alternatively, $|\hat{\phi}_{exp}[n]| \propto \alpha/(\alpha^2 + n^2)$, indicating that the frequency amplitudes follow Lorentzian decay controlled by the decay rate $\alpha$. Lorentzian decay is less aggressive in reducing high-frequency components, which helps preserve fine details and sharp transitions in the signal. This flexibility helps tune the decay based on signal characteristics. However, the slower decay rate may lead to less effective noise suppression.

We can now look at the frequency spectrum of the full surrogate model $u_\theta(x)$ in Equation 8. We use the convolution theorem (McGillem & Cooper, 1991) to obtain

$$\hat{u}_\theta(\omega) = \hat{\phi} * \hat{u}_N = \int_{-\infty}^{\infty} \hat{\phi}(\omega - \nu)\, \hat{u}_N(\nu)\, \mathrm{d}\nu \tag{22}$$

where $*$ denotes convolution, and $\hat{u}_N$ is the Fourier transform of the NN. Suppose $\omega > 0$, we know that $\hat{\phi}(\omega - \nu)$ is obtained by first reflecting $\hat{\phi}(\nu)$ about the y-axis, and then shifting the frequency $\omega$ left along the spectrum. Then, we integrate $\hat{u}_N(\nu)$ weighted by $\hat{\phi}(\omega - \nu)$. The larger the frequency $\omega$, the more $\hat{\phi}$ is moved left to obtain $\hat{\phi}(\omega - \nu)$ resulting in a stronger weighting of the high-frequency components during integration. Since the frequency coefficients of $\hat{\phi}$ decay quadratically fast ($|\hat{\phi}[n]| \propto 1/n^2$), the larger the portion of the tail is used, and the smaller the integration result. Specifically, with $\phi$, the corresponding amplitude of $\hat{u}_\theta(\omega)$ decrease fast when $\omega$ increases[2]. See Figure 3 for an illustration in the discrete case. During training, the boundary function pushes the surrogate model in Equation 8 to weaken irrelevant high-frequency components to reduce large tails and better capture the frequency spectrum.

---

[1]While the analysis here is presented in the discrete Fourier series form for clarity and consistency with the numerical implementation, the results can be extended to the continuous Fourier transform by replacing summations over discrete indices $n$ with integrals over the continuous frequency domain.

[2]The same conclusion applies when we consider $\omega < 0$.

## 5 Fourier PINNs

Our Fourier analysis in the previous section revealed that the standard PINNs inefficiently capture the true amplitude of frequencies in the power spectrum of their predicted solutions. Specifically, the analysis showed that although standard PINNs could identify the true solution frequency, they failed to prune large and noisy frequencies, as indicated by the large amplitudes assigned at the tail end of the power spectrum in Figure 3. While the strong BC PINNs address this issue by multiplying the NN with a boundary function—which helps decay high-frequency amplitudes faster—their overall benefit remains limited. As shown in Figure 2, performance degrades as $k$ increases, indicating the boundary function's inability to learn ultra-high frequency solutions. The strong BC PINNs are also restricted to specific boundary conditions, such as Dirichlet conditions in Equation 7, limiting their applicability.

### 5.1 Fourier PINNs

We introduce Fourier PINNs, a novel PINN architecture to overcome these challenges. Fourier PINNs seamlessly handle diverse boundary conditions, like the standard PINNs, while emphasizing the true solution frequencies during training—akin to the strong BC PINNs. Specifically, in order to flexibly yet comprehensively capture the frequency spectrum, we first introduce a set of dense frequency candidates, $\{\omega_n\}_{n=1}^K$, evenly sampled from the range $[1, K]$. From this set of frequency candidates, we define a set of trainable Fourier bases $u_B$ as,

$$u_B(\mathbf{x}; \mathbf{a}, \mathbf{b}) = \sum_{n=1}^K a_n \cos(2\pi\omega_n\mathbf{x}) + b_n \sin(2\pi\omega_n\mathbf{x}). \tag{23}$$

The Fourier PINN architecture additively combines the outputs of both the NN from Equation 3 and the Fourier bases from Equation 23 to generate the augmented prediction $u_F$. Specifically, the full output of Fourier PINNs is,

$$u_F(\mathbf{x}; \boldsymbol{\theta}) = \sum_{j=1}^W c_j \psi_j(\mathbf{x}; \boldsymbol{\theta}^H) + \sum_{n=1}^K a_n \cos(2\pi\omega_n\mathbf{x}) + b_n \sin(2\pi\omega_n\mathbf{x}). \tag{24}$$

We define $\mathbf{w} = \{c_1, ..., c_W, a_1, ..., a_K, b_1, ..., b_K\}$ as the set of all basis coefficients, and $\boldsymbol{\theta} = \{\boldsymbol{\theta}^H, \mathbf{w}\}$ form the set of all trainable network parameters.

While we present our method in the one-dimensional input case, extending it to multiple dimensions is straightforward. Consider the two-dimensional input $\mathbf{x} = [x_1, x_2]$ for an example. We use the same set of frequencies to construct Fourier bases for each input. Specifically, we construct

$$\boldsymbol{\phi}(x_1) = [\cos(2\pi\omega_1 x_1), \ \sin(2\pi\omega_1 x_1), \ \ldots, \ \cos(2\pi\omega_K x_1), \ \sin(2\pi\omega_K x_1)],$$

and

$$\boldsymbol{\phi}(x_2) = [\cos(2\pi\omega_1 x_2), \ \sin(2\pi\omega_1 x_2), \ \ldots, \ \cos(2\pi\omega_K x_2), \ \sin(2\pi\omega_K x_2)].$$

Then, we apply the cross-product of $\boldsymbol{\phi}(x_1)$ and $\boldsymbol{\phi}(x_2)$ to obtain the tensor-product Fourier bases for $\mathbf{x}$. The surrogate model is given by,

$$u_{\boldsymbol{\theta}}(\mathbf{x}) = u_N(\mathbf{x}) + \boldsymbol{\beta}^\top \text{vec}\left(\boldsymbol{\phi}(x_1)\boldsymbol{\phi}(x_2)^\top\right) \tag{25}$$

where $\text{vec}(\cdot)$ denotes the vectorization, and $\boldsymbol{\beta}$ are the coefficients of the Fourier bases. Note that the tensor-product of bases grows exponentially with the problem dimension, quickly becoming computationally intractable. One potential solution is to generate more sparse Fourier bases, such as total-degree or hyperbolic cross bases. Another option is to re-organize the coefficients $\boldsymbol{\beta}$ into a tensor or matrix and introduce a low-rank decomposition structure to parameterize $\boldsymbol{\beta}$ to save computational cost (Novikov et al., 2015). These explorations are left for future work.

## 5.2 Adaptive Basis Selection Training Algorithm

Given that the ground-truth solution likely contains significantly fewer frequencies than the augmented candidate bases ($u_B$), and our previous analysis showed that models often struggle to prioritize learning necessary frequencies, we developed an adaptive learning and basis selection algorithm. This algorithm flexibly identifies meaningful frequencies while pruning the inconsequential ones—allowing the model to focus on learning and improving the bases that significantly contribute to the solution by inhibiting the learning of unnecessary (and likely noisy) frequencies.

To aid in identifying the significant frequencies, we add an $L_2$ regularization term to the basis parameters $\mathbf{w}$. While $L_2$ regularization does not promote sparsity, in our specific case, we use $L_2$ regularization with a pruning strategy, which indirectly promotes sparsity by pruning coefficients after driving them toward a threshold. Methods such as the SINDy (Sparse Identification of Nonlinear Dynamics) algorithm Zhang & Schaeffer (2019) have explored this strategy, where $L_2$ regularization helps to stabilize the coefficients during optimization before pruning—yielding sparse solutions. This combination effectively balances stability and sparsity, as shown in previous work Wang et al. (2021a). In this case, we chose not to use $L_1$ regularization due to practical challenges, including the instability it can introduce during training, particularly when optimizing neural networks for complex systems. $L_2$ regularization, combined with a careful pruning process, offers more stable convergence while still leading to sparse solutions after pruning the small coefficients.

This additional regularization penalizes large coefficients for the basis functions, thereby promoting sparsity and encouraging the model to focus on learning the most significant frequencies. The full loss function we use is similar to PINNs (see Equation 4) but with the addition of the $L_2$ regularization term. Specifically, the loss function is,

$$L(\boldsymbol{\theta}) = L_b(\boldsymbol{\theta}) + L_r(\boldsymbol{\theta}) + \frac{\alpha}{2} \|\mathbf{w}\|^2, \tag{26}$$

where $\|\mathbf{w}\|^2$ represents the $L_2$ regularization of $\mathbf{w}$ and $\alpha$ is the regularization strength defined by the user before initiating training. This regularization term helps prune or inhibit unnecessary frequencies, allowing the model to focus more on learning and improving the bases that significantly contribute to the solution.

**Linear Operators** Our adaptive basis selection algorithm optimizes the regularized loss function in Equation 26 by building upon the hybrid least squares gradient descent method proposed by Cyr et al. (2020). Cyr et al. (2020) formulates the output of a neural network as a linear combination of nonlinear basis functions, as shown in Equation 3. Their algorithm alternates between optimizing the hidden weights ($\boldsymbol{\theta}^H$) via gradient descent and the final layer coefficients ($\mathbf{c}$) of the NN through the least squares problem of the form,

$$\arg\min_{\mathbf{c}} \|A\mathbf{c} - \mathbf{y}\|_{\ell_2}^2, \tag{27}$$

such that $\mathbf{y}_i = \mathcal{L}[u](\mathbf{x}_i)$ for the data points $\{\mathbf{x}_i\}_{i=1}^M$, $A_{ij} = \mathcal{L}[\psi_j(\mathbf{x}_i; \boldsymbol{\theta}^H)]$ for $j = 1, ..., W$, and $\mathcal{L}$ is some linear operator. Equation 27 extends to problems with multiple operators, such as those defined by Equation 1 and Equation 2 when $\mathcal{F}$ and $\mathcal{B}$ are linear operators.

Representing the output of Fourier PINNs in Equation 23 as a linear combination of both adaptive bases (i.e., the hidden layers of the NN) and an augmented Fourier series ensures our model's compatibility with the general approach of the alternating least squares and gradient descent optimization. While Cyr et al. (2020) hold the coefficients of the last layer in the NN ($\mathbf{c}$) constant while optimizing the bases $\boldsymbol{\theta}^H$ through gradient descent, we alternatively choose to jointly optimize both the Fourier PINN's basis coefficients $\mathbf{w}$ (which are comprised of both the NN basis coefficients $\mathbf{c}$ along with the Fourier basis coefficients $a_1, ..., a_K$ and $b_1, ..., b_K$) and the adaptive bases $\boldsymbol{\theta}^H$ during the gradient descent step. During the least squares step, we solve a similar problem to Equation 27 modified to suit the Fourier PINN architecture. Specifically, we solve the least squares problem of the form,

$$\arg\min_{\mathbf{w}} \|A'\mathbf{w} - \mathbf{y}\|_{\ell_2}^2, \tag{28}$$

such that $\mathbf{y}_i = \mathcal{L}[u](\mathbf{x}_i)$ for the data points $\{\mathbf{x}_i\}_{i=1}^M$, and

$$A'_{ij} = \begin{cases} \mathcal{L}[\psi_j(\mathbf{x}_i; \boldsymbol{\theta}^H)] & \text{for } j = 1, ..., W, \\ \mathcal{L}[\cos(2\pi\omega_j\mathbf{x}_i)] & \text{for } j = W+1, ..., W+K, \\ \mathcal{L}[\sin(2\pi\omega_j\mathbf{x}_i)] & \text{for } j = W+K+2, ..., W+2K. \end{cases} \tag{29}$$

We define Equation 28 for one operator $\mathcal{L}$, but the method extends to multiple operators. Additionally, we extend our least-squares method to nonlinear operators, unlike many works that focus only on linear cases.

**Non-Linear Operators** When the operator $\mathcal{L}$ is nonlinear, updating the coefficients $\mathbf{w}$ requires additional care to ensure the method accurately handles the nonlinearities and achieves convergence. Nonlinear operators can often be decomposed into a combination of linear and nonlinear components, simplifying the process of updating $\mathbf{w}$ during optimization. Specifically, we decompose the operator $\mathcal{L}$ as follows:

$$\mathcal{L}[\cdot] = \mathcal{G}[\cdot] + \mathcal{S}[\cdot] \tag{30}$$

where $\mathcal{G}$ represents the linear part of the operator, and $\mathcal{S}$ denotes the nonlinear component.

To illustrate this, consider the Allen-Cahn equation, where the operator $\mathcal{L}$ contains both linear and nonlinear terms. In this case, the linear part $\mathcal{G}$ is $u_{xx}$, and the nonlinear term $\mathcal{S}[u]$ is $u(u^2 - 1)$, leading to the decomposition:

$$\mathcal{G}[u] = u_{xx}, \quad \mathcal{S}[u] = u(u^2 - 1). \tag{31}$$

We solve this system by incorporating the Fourier PINN formulation. For the least squares solution of the system, the nonlinear operator $\mathcal{S}$ is applied at each iteration using the current approximation of $u$, which is based on the current coefficients $\mathbf{w}$ and is given as,

$$\mathbf{s}_i = \mathcal{S}[u_F](\mathbf{x}_i) = u_F(\mathbf{x}_i; \mathbf{w})(u_F(\mathbf{x}_i; \mathbf{w})^2 - 1) \tag{32}$$

Here, $u_F(\mathbf{x}_i; \mathbf{w})$ is the Fourier PINN's evaluation at point $\mathbf{x}_i$, using the current parameters $\mathbf{w}$. The linear part $\mathcal{G}$ is handled directly in the least squares system, and the matrix $A'$ is defined as:

$$A'_{ij} = \begin{cases} \mathcal{G}[\psi_j(\mathbf{x}_i; \boldsymbol{\theta}^H)] & \text{for } j = 1, ..., W, \\ \mathcal{G}[\cos(2\pi\omega_j\mathbf{x}_i)] & \text{for } j = W+1, ..., W+K, \\ \mathcal{G}[\sin(2\pi\omega_j\mathbf{x}_i)] & \text{for } j = W+K+2, ..., W+2K. \end{cases} \tag{33}$$

The nonlinear terms $\mathbf{s}_i$ are included in the target vector such that $\mathbf{y}_i = \mathcal{L}[u](\mathbf{x}_i) - \mathbf{s}_i$, which is updated iteratively and reduces the problem to a linear least-squares system. As this method incorporates the current values of $\mathbf{w}$ into calculating the nonlinear components, the optimization process can be viewed as a form of fixed-point iteration. We note that for cases where $\mathcal{L}$ consists primarily of nonlinear terms or does not include a linear operator, the optimization problem might need to be handled using continuous optimization algorithms such as L-BFGS, which effectively updates $\mathbf{w}$ based on the entire system to ensure convergence.

We integrate a custom basis pruning routine into the optimization algorithm to eliminate insignificant bases and refine model training. Our modified loss function defined in Equation 26 enhances this hybrid optimization algorithm with an $L_2$ regularization term to identify and prioritize significant frequencies. Algorithm 1 outlines our adaptive learning routine. The algorithm begins with a joint optimization of $\boldsymbol{\theta}^H$ and $\mathbf{w}$ to provide a warm start to the model parameters. Next, we fix $\boldsymbol{\theta}^H$ while alternating between solving for the coefficients $\mathbf{w}$ using the least squares method and truncating bases with coefficients below a predefined threshold. Specifically, bases corresponding to $|w_j| \leq 10^{-3}$ are removed along with $w_j$. After completing the alternating solve and truncating steps for a set number of iterations, the algorithm jointly optimizes all remaining parameters and bases using the Adam optimization algorithm. We repeat this routine for a fixed number of iterations, followed by L-BFGS optimization until the final convergence.

---

**Algorithm 1:** Adaptive Basis Selection Hybrid Least Squares/Gradient Descent

---

**Input:** $\boldsymbol{\theta}_0^H$ *(initialized hidden parameters)*
**Output:** Optimized parameters $\boldsymbol{\theta}^H$ and weights $\boldsymbol{w}$
$\boldsymbol{w} \leftarrow \text{LS}(\boldsymbol{\theta}_0^H)$ ;                                        // Solve LS problem for $\boldsymbol{w}$
$\boldsymbol{\theta}^H, \boldsymbol{w} \leftarrow \text{ADAM}(\boldsymbol{\theta}_0^H, \boldsymbol{w})$ ;                              // Initialize $\boldsymbol{\theta}^H$ and $\boldsymbol{w}$
**for** $i = 1, \dots$ **do**
    **for** $k = 1, \dots$ **do**
        $\boldsymbol{w} \leftarrow \text{LS}(\boldsymbol{\theta}^H)$ ;                                   // Solve LS problem for $\boldsymbol{w}$
        **if** $|w_j| < \delta$ *for each $w_j$* **then**
            Prune the corresponding basis;
            Delete $w_j$ from $\boldsymbol{w}$;
        **end**
    **end**
    $\boldsymbol{\theta}^H, \boldsymbol{w} \leftarrow \text{ADAM}(\boldsymbol{\theta}^H, \boldsymbol{w})$ ;            // Jointly optimize all parameters using ADAM
**end**
Run L-BFGS until convergence;

---

## 6 Experiments

This section comprehensively evaluates our methods across a diverse set of partial differential equation (PDE) problems. The test cases include the 1D and 2D Poisson equations, which serve as canonical examples of linear elliptic PDEs, and the 1D and 2D steady-state Allen-Cahn equations, representing nonlinear elliptic PDEs with a nonlinear reaction term. To further challenge the models, we analyze the 1D one-way Wave equation, which incorporates a first-order time derivative, and the 1D non-steady-state Allen-Cahn equation, a stiff PDE with non-harmonic characteristics. We first describe the baseline methods for comparing Fourier PINNs and then outline the experimental setup. Finally, we present and discuss the results, highlighting the relative performance of each method on the benchmark problems.

### 6.1 Baselines

We benchmarked against the following state-of-the-art PINN models for solving high-frequency and multi-scale solutions:

(1) (PINN) standard PINNs as formulated in (Raissi et al., 2017);

(2) (RFF-PINN) Random Fourier Feature PINNs (Wang et al., 2021c) with dynamically re-weighted loss terms derived from the NTK eigenvalues as in Wang et al. (2022),

(3) (W-PINN) Weighted PINNs that down-weight the residual loss term to reduce its dominance and to fit the boundary loss Wang et al. (2022) better;

(4) (A-PINN) Adaptive PINNs with parameterized activation functions to increase the NN capacity and to be less prone to gradient vanishing and exploding Jagtap et al. (2020); and

(5) (Spectral) A single Fourier layer comprised of Fourier bases and trainable coefficients. Basis coefficients are optimized through the same gradient descent routines as all other baselines[3].

We do not test against the conceptually similar Fourier Neural Operators (FNOs) (Li et al., 2021), as they are designed for inverse problems and rely on a data loss term. In contrast, our method focuses on solving forward problems and does not require training data within the domain. Additionally, while the methods listed above represent widely-used PINN architectures, we acknowledge the existence of alternative neural

---

[3]Throughout the experiments, we used the same set of Fourier bases for the spectral method and Fourier PINNs.

network approaches based on random features and random projections, such as Random Vector Functional Link networks (RVFLNs) (Zhang & Suganthan, 2016), Extreme Learning Machines (ELMs) (Ding et al., 2013; Fabiani et al., 2021; Calabrò et al., 2021), Random Projection Neural Networks (RPNNs) (Rahimi & Recht, 2008), Reservoir Computing (Pathak et al., 2018), and recent advancements like RandONet (Fabiani et al., 2025). While methods such as RandONets and RVFLNs demonstrate significant computational efficiency and excel in inverse problem-solving tasks, their ability to address spectral bias in forward high-frequency problems remains underexplored.

By contrast, the proposed Fourier PINN architecture directly targets spectral bias—a key limitation of existing PINN approaches in learning high-frequency and multi-scale solutions. Unlike conventional architectures, the Fourier PINN framework is both modular and adaptable, with the integration of Fourier bases capable of enhancing or complementing any of these alternative methods. Given its flexibility and minimal downsides, exploring how Fourier PINNs could be combined with these approaches to improve scalability and accuracy remains an exciting direction for future work.

Table 2: *The number and corresponding scales of Gaussian variances utilized in the RFF-PINN. Scales were selected randomly from a set range of* $[1, 200]$. *Preference was given to the subset* $[1, 20, 50, 100]$ *to ensure a balanced distribution across the available range.*

| Number | Scales |
|---|---|
| 1 | $(20)$, $(50)$, $(84)$, $(100)$ |
| 2 | $(1, 50)$, $(3, 20)$, $(19, 71)$, $(39, 69)$, $(50, 100)$ |
| 3 | $(44, 47, 165)$, $(1, 20, 194)$, $(20, 50, 100)$, $(1, 50, 189)$, $(38, 112, 119)$ |
| 5 | $(1, 20, 49, 50, 100)$, $(1, 20, 50, 85, 100)$ |
| | $(1, 20, 104, 197, 199)$, $(6, 36, 67, 79, 136)$, $(50, 65, 83, 104, 139)$ |

## 6.2 Hyperparameters and experimental details

For each experiment, we trained the models in two stages: first using the Adam optimizer (Kingma & Ba, 2015), followed by L-BFGS optimization (Liu & Nocedal, 1989) until convergence, with the tolerance set to $10^{-9}$. Specific hyperparameters and training details are provided in the following tables: Table 5 for the 1D Poisson and 1D Allen-Cahn experiments, Table 6 for the 1D Wave equation experiments, Table 7 for the 2D Poisson and 2D (steady-state) Allen-Cahn experiments, and Table 8 for the 1D (non-steady-state) Allen-Cahn experiment. All experiments use 32-bit floating-point precision (float32) to ensure computational efficiency and consistency across models and were conducted on a GeForce RTX 3090 GPU with CUDA version 12.3, running on Ubuntu 20.04.6 LTS. In Fourier PINNs, we determined the range of frequency candidates for the Fourier bases by setting a maximum frequency $K$ and including all equally spaced frequencies in the set $\{1, 2, ..., K\}$. For the alternating optimization algorithm (Algorithm 1), we set the basis truncation threshold $\delta = 10^{-4}$ and the regularization strength parameter $\alpha = 10^{-4}$. Optimization used the same number of Adam iterations across experiments, with pauses at every 1000th iteration, to solve the LS problem for five iterations. RFF-PINNs required specifying the number and scales of Gaussian variances used to construct the random features. To evaluate the effects of these hyperparameters, we tested 20 different settings, varying the number of scales (one, two, three, and five) and using variances suggested in (Wang et al., 2021c) along with randomly sampled values. Table 2 provides the exact sets of scales. For W-PINN, we varied the weight of the residual loss from $\{10^{-1}, 10^{-3}, 10^{-4}\}$ and reported the configuration achieving the best results. For A-PINN, we introduced a learned parameter for the activation function in each layer and updated these parameters jointly. We ran every method for five random trials and reported the five number summaries obtained on seperate testing datasets for each experiment in boxplot form. All code is implemented using the PyTorch C++ library Paszke et al. (2019)[4].

---

[4]Code is available at https://github.com/VarShankar/KernelPack/tree/sciml

### 6.3 1D Numerical Experiments

This section evaluates Fourier PINNs on three progressively challenging 1D Poisson and Allen-Cahn benchmarks, demonstrating their superior performance and robustness over the baseline methods. These benchmarks systematically assess the models' ability to handle high-frequency, multi-scale, and hybrid-frequency solutions for linear and non-linear elliptic PDEs. The analysis includes relative $\ell_2$ error comparisons, convergence dynamics, and frequency spectrum decomposition to highlight the strengths of Fourier PINNs. Table 5 in the Appendix lists each example's hyperparameters and experimental details.

#### 6.3.1 1D Poisson Problem

The 1D Poisson problem evaluates the ability of Fourier PINNs to capture high-frequency signals within a linear context. We test the following manufactured solutions with increasing frequency complexity:

$$u(x) = \sin(100x), \tag{34}$$

$$u(x) = \sin(x) + 0.1\sin(20x) + 0.05\cos(100x), \tag{35}$$

$$u(x) = \sin(6x)\cos(100x). \tag{36}$$

The boundary conditions are derived directly from the true solutions. These tests progressively introduce multi-scale and hybrid frequency components used to study the robustness and scalability of each model.

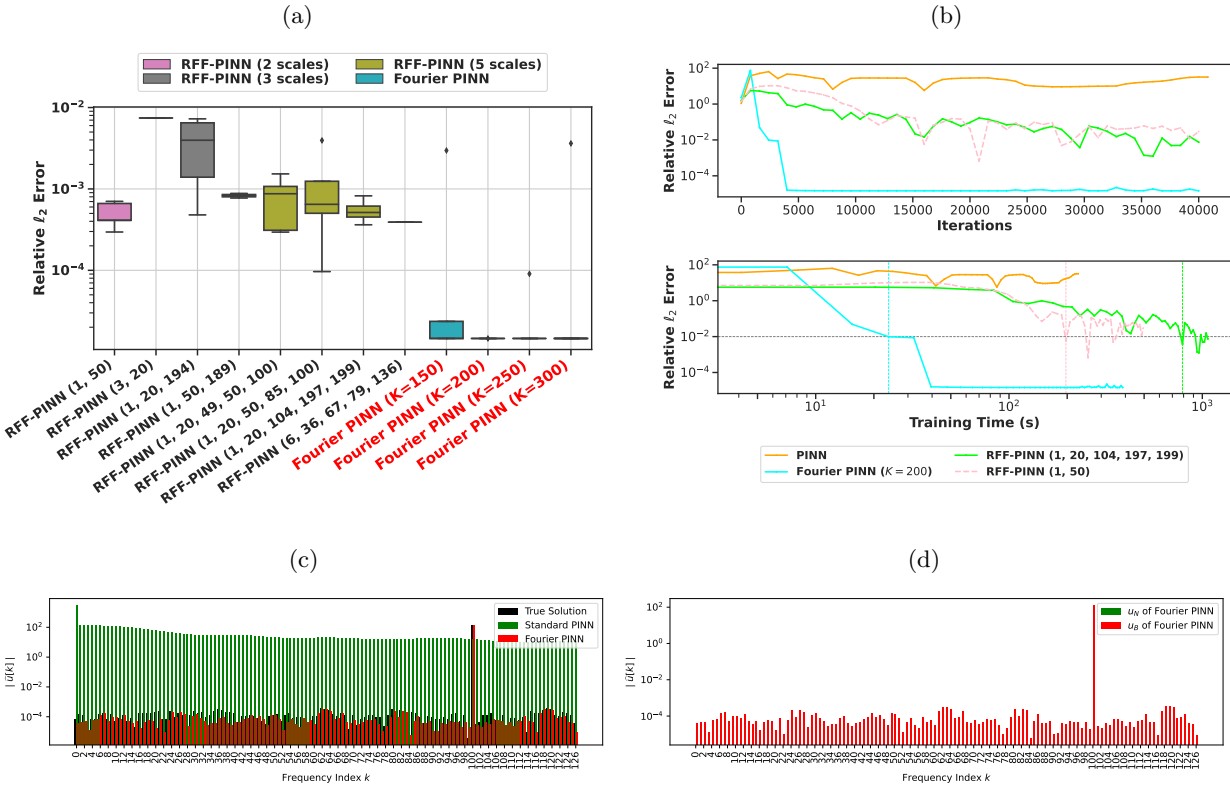

Figure 7: *Results for the single-scale true solution $u(x) = \sin(100x)$. (a) Boxplot showing the relative $\ell_2$ errors for Fourier PINNs and the top-performing baseline methods. (b) Convergence plots comparing the relative $\ell_2$ error across iterations (top) and training time (bottom) for selected models. Vertical lines indicate when each model reached an error of $10^{-2}$. (c) Frequency spectrum of the solution $u(x)$ compared to PINN and Fourier PINN approximations. (d) Decomposition of the frequency components in Fourier PINNs between the NN output and the Fourier layer output, highlighting differences in spectral representation.*

**Single-Scale Solution Results.** Figure 7a shows the relative $\ell_2$ error distributions for Fourier PINNs and top baseline models, while Figure 15a in the Appendix includes results for all models. Fourier PINNs

consistently achieve significantly lower errors than baselines, including RFF-PINNs and spectral methods. Notably, RFF-PINNs demonstrate sensitivity to the chosen scale configurations, which impacts their overall performance. Figure 7b illustrates the convergence behavior across iterations and training time to analyze training dynamics. Fourier PINNs exhibit rapid and stable convergence due to the alternating optimization and basis truncation routine. In contrast, standard PINNs plateau at higher errors, struggling to approximate the high-frequency features of the solution accurately.

As shown in Figure 7c, the frequency domain analysis compares the Discrete Fourier Transform (DFT) of the solutions from standard PINNs and Fourier PINNs against the ground truth. Fourier PINNs generate frequency spectra almost identical to the ground truth, effectively capturing the high-frequency features. Conversely, standard PINNs fail to resolve the high-frequency components, underscoring their limitations for high-frequency problems. To further dissect Fourier PINN solutions, we decomposed the predicted solution, $u_F = u_N + u_B$, into its neural network (NN) and Fourier layer components. Figure 7d presents the frequency spectra of each component. The absence of green bars (representing the NN component) indicates no contribution from the NN basis. This result aligns with the basis truncation outcome, where all NN bases were truncated during training, leaving only a single Fourier basis. This observation confirms that the optimization algorithm accurately captures the frequency components of the true solution.

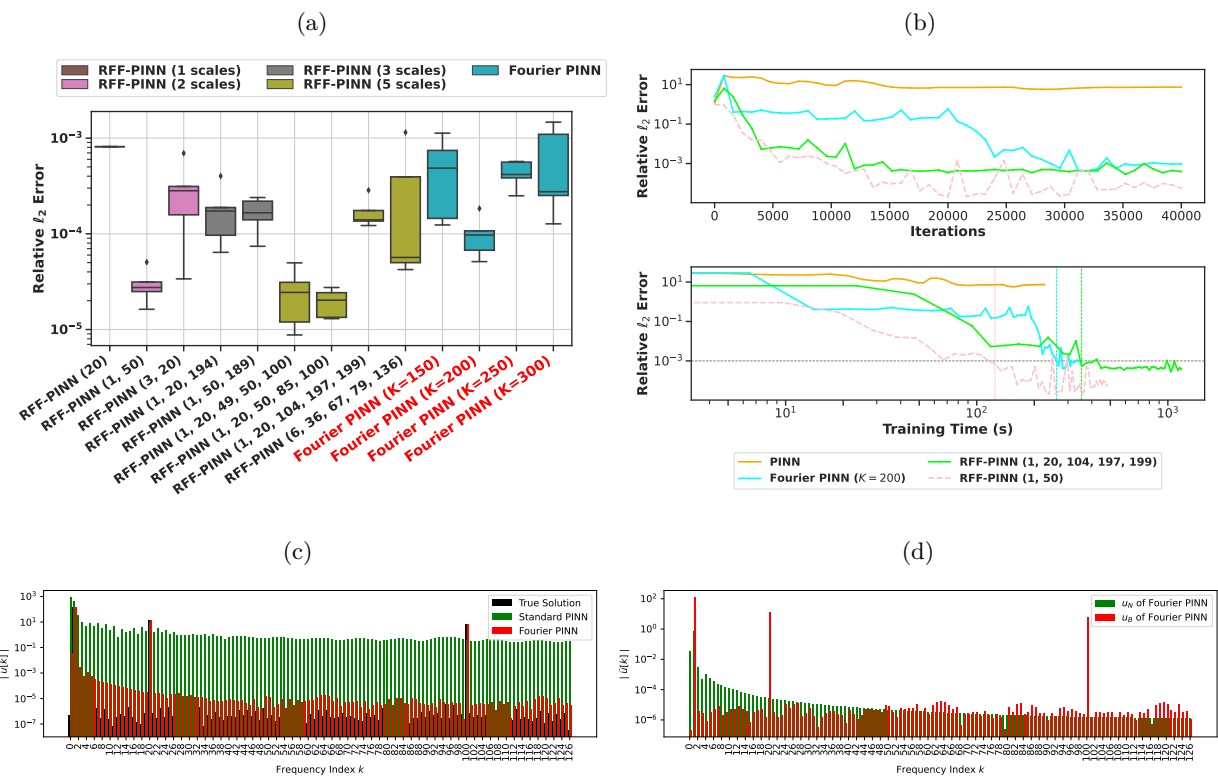

Figure 8: *(1D Poisson) Results for the multi-scale true solution $u(x) = \sin(x) + 0.1\sin(20x) + 0.05\cos(100x)$. (a) Boxplot showing the relative $\ell_2$ errors for Fourier PINNs and the top-performing baseline methods. (b) Convergence plots comparing the relative $\ell_2$ error across iterations (top) and training time (bottom) for selected models. Vertical lines indicate when each model reached an error of $10^{-3}$. (c) Frequency spectrum of the solution $u(x)$ compared to PINN and Fourier PINN approximations. (d) Decomposition of the frequency components in Fourier PINNs between the NN output and the Fourier layer output, highlighting differences in spectral representation.*

**Multi-Scale Solution Results.** The multi-scale problem introduces a combination of low- and high-frequency components, presenting a greater challenge for baseline methods. Figure 8a shows the relative $\ell_2$ error distributions for Fourier PINNs and top baseline models, while Figure 15b in the Appendix includes results for all models. Fourier PINNs outperform most baseline methods; however, a subset of RFF-PINN configurations achieves slightly better results. These cases highlight the sensitivity of RFF-PINNs to scale

selection, whereas Fourier PINNs demonstrate robust performance across all tested basis configurations. RFF-PINNs exhibit more significant variability across configurations, with inconsistent performance due to their reliance on precise scale tuning.

The frequency spectrum comparison in Figure 8c illustrates that Fourier PINNs accurately reconstruct both low- and high-frequency components, highlighting their adaptability to multi-scale structures. In contrast, the spectrum of the standard PINN output shows that it accurately captures low-frequency components but struggles to resolve the solution's high-frequency features. Figure 8d further analyzes the frequency decomposition of the Fourier PINN solution. The Fourier layer captures all high-frequency components of the target solution (e.g., $\sin(20x)$ and $\cos(100x)$ in Equation 35), while the neural network (NN) output contributes to modeling the low-frequency component ($\sin(x)$). This result aligns with the basis truncation outcomes: three Fourier bases were retained, and approximately 10% of the NN basis functions were truncated during training. These findings demonstrate that the hybrid architecture of Fourier PINNs, combined with the alternating optimization and truncation algorithm, effectively balances the modeling of both smooth and oscillatory components in multi-scale solutions.

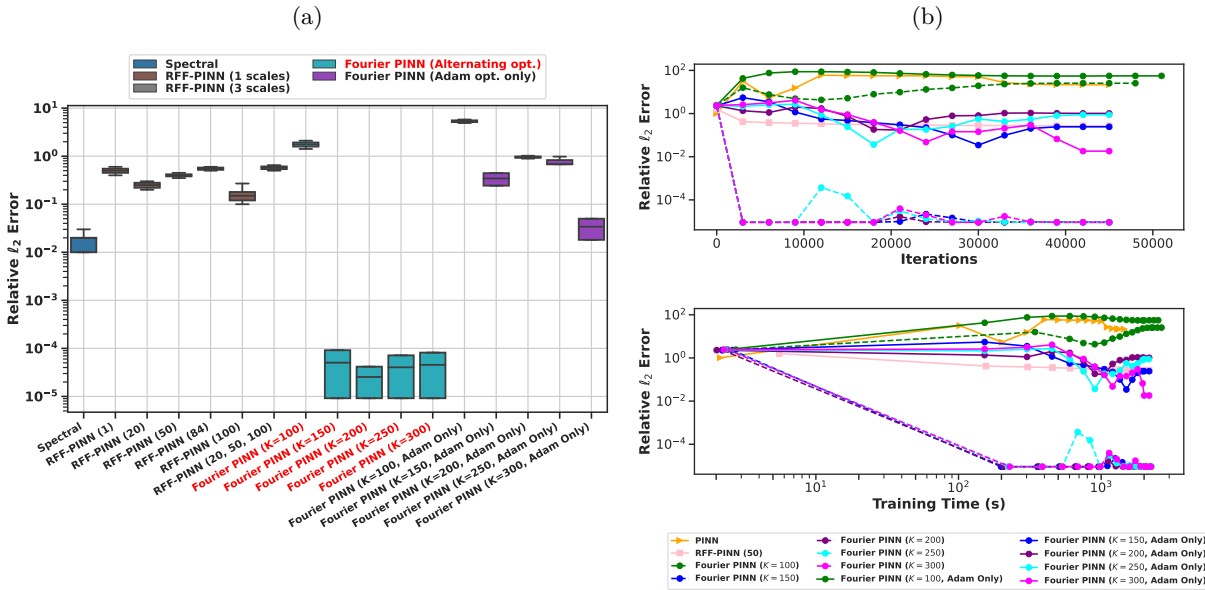

Figure 9: *(1D Poisson) Results for the single-scale true solution $u(x) = \sin(6x)\cos(100x)$. (a) Boxplot showing the relative $\ell_2$ errors for Fourier PINNs and the top-performing baseline methods. (b) Convergence plots comparing the relative $\ell_2$ error across iterations (top) and training time (bottom) for selected models.*

**Hybrid Solution Results.** This hybrid-frequency example introduces closely spaced high-frequency components, which increases the complexity of disentangling and modeling each frequency. Figure 9a illustrates the relative $\ell_2$ error distributions for Fourier PINNs and top baseline models, while Figure 15c in the Appendix presents results for all models. To specifically evaluate the impact of the alternating optimization algorithm on Fourier PINNs, we tested multiple basis configurations using the alternating optimization method described in Algorithm 1, as well as traditional optimization routines (i.e., without the least-squares (LS) basis coefficient optimization).

The results demonstrate that Fourier PINNs, when combined with the alternating optimization strategy, consistently achieve the lowest $\ell_2$ errors across all tested configurations, highlighting their ability to effectively model solutions with coupled and closely spaced frequency components. In contrast, Fourier PINNs without alternating optimization fail to accurately capture the true solution, emphasizing the critical role of the LS basis coefficient updates in resolving hybrid-frequency structures. The convergence behavior shown in Figure 9b further underscores the efficiency of Fourier PINNs in learning hybrid solutions. Their rapid convergence and stable performance across iterations showcase their robustness and adaptability to challeng-

ing problem setups with complex frequency interactions. Additionally, there is little variation in training time with increasing Fourier basis sizes.

### 6.3.2 1D Steady-State Allen-Cahn Problem

Next, we examine the 1D steady-state Allen-Cahn equation, which is a nonlinear reaction-diffusion system. This problem serves as a more complex PDE benchmark for evaluating Fourier PINNs and baseline models. To ensure consistency, we test the same manufactured solutions used in the 1D Poisson case, defined by Equation 34, Equation 35, and Equation 36.

**Results.** Figure 16 presents the relative $\ell_2$ error distributions for Fourier PINNs and the top-performing baseline models across all manufactured solutions, while detailed results for all models can be found in Figure 16 in the Appendix. Across all configurations, Fourier PINNs consistently achieve lower relative $\ell_2$ errors. Additionally, their faster convergence, as highlighted in Figure 11, underscores the efficiency of the Fourier PINN framework for solving the 1D steady-state Allen-Cahn problem.

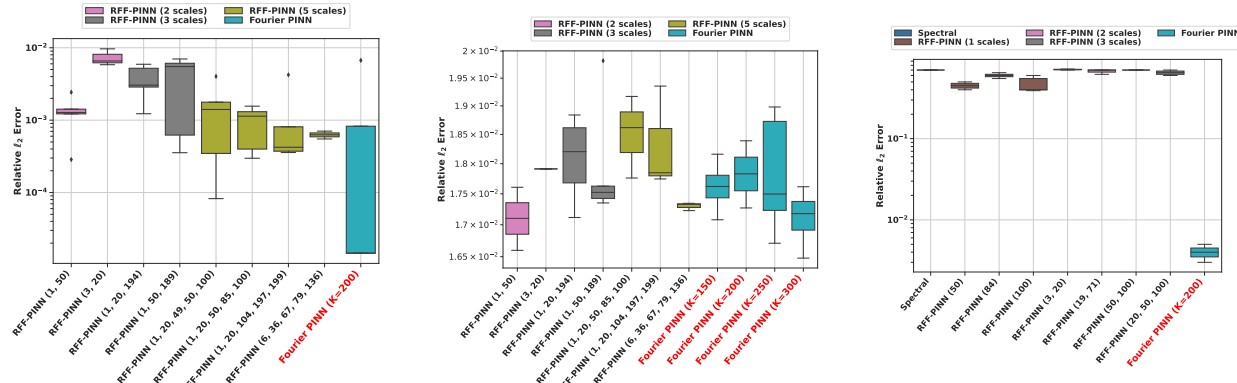

Figure 10: *(1D Allen-Cahn) Boxplot of relative $\ell_2$ errors for Fourier PINNs and top-performing baseline methods across the manufactured solutions: left, $u(x) = \sin(100x)$; middle, $u(x) = \sin(x) + 0.1\sin(20x) + 0.05\cos(100x)$; and right, $u(x) = \sin(6x)\cos(100x)$.*

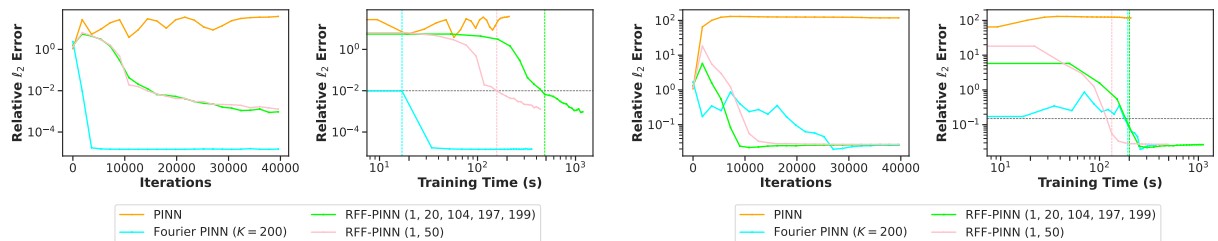

Figure 11: *Convergence behavior of select models on the 1D steady-state Allen-Cahn problems, showing the mean relative $\ell_2$ error as a function of iterations (left) and training time (right). Results are presented for $u(x) = \sin(100x)$ (left two plots) and $u(x) = \sin(x) + 0.1\sin(20x) + 0.05\cos(100x)$ (right two plots).*

### 6.4 2D Numerical Experiments

We evaluate Fourier PINNs on two 2D PDEs: the 2D Poisson problem and the steady-state Allen-Cahn equation. These examples test the model's scalability, ability to capture high-frequency components, and approximation of multi-scale solutions in higher dimensions. Table 7 in the Appendix lists each example's hyperparameters and experimental details.

### 6.4.1 2D Poisson Problem

The 2D Poisson problem is given by:

$$\Delta u = f(x, y), \quad x, y \in [0, 2\pi], \tag{37}$$

with the following ground-truth solutions:

$$u(x, y) = \sin(100x)\sin(100y), \tag{38}$$

$$u(x, y) = \sin(6x)\cos(20x) + \sin(6y)\cos(20y). \tag{39}$$

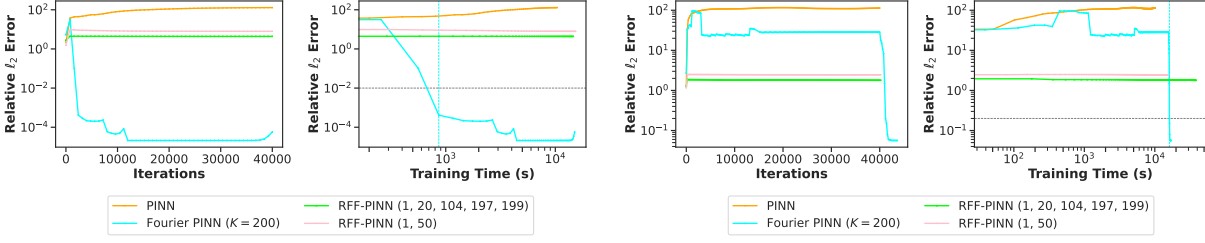

Figure 12: *Convergence behavior of select models on the 2D Poisson problems, showing the mean relative $\ell_2$ error as a function of iterations (left) and training time (right). Results are presented for Equation 38 (left two plots) and Equation 39 (right two plots).*

**Results.** For the high-frequency solution (Equation 38), Fourier PINNs achieve an average relative $\ell_2$ error of 0.0384, outperforming all baseline methods, with the spectral method being the closest competitor at 0.0423. The Appendix provides detailed results for all models in Figure 17. Notably, other methods fail to capture the high-frequency features, with errors exceeding 1. For the multi-scale solution (Equation 39), Fourier PINNs demonstrate a significant advantage, achieving an average relative $\ell_2$ error of 0.00085, far surpassing the performance of all other baselines. Figure 12 highlights the superior convergence behavior of Fourier PINNs in both scenarios, underscoring their ability to handle challenging high-frequency and multi-scale problems effectively.

### 6.4.2 2D Steady-State Allen-Cahn Equation

The Allen-Cahn problem is defined as:

$$u_{xx} + u_{yy} + u(u^2 - 1) = f(x, y), \quad x, y \in [0, 2\pi], \tag{40}$$

with the ground-truth solution:

$$u(x, y) = (\sin(x) + 0.1\sin(20x) + \cos(100x)) \cdot (\sin(y) + 0.1\sin(20y) + \cos(100y)). \tag{41}$$

**Results:** The Allen-Cahn example highlights Fourier PINNs' challenges with nonlinear PDEs. The model achieves a relative $\ell_2$ error of 0.9, outperforming all other methods ($> 1$). In the Appendix, Figure 18 provides detailed results for all models. This suggests the need for further optimization for highly nonlinear problems.

### 6.5 1D One-Way Wave Problem

We next tested the 1D One-Way Wave equation of the form,

$$u_t + 10u_x = v(x, t), \quad x \in [0, 1], t \in [0, 1] \tag{42}$$
$$u(x = 0, t) = u(x = 1, t) = 0$$
$$u(x, t = 0) = \sin(\pi x) + \sin(2\pi x)$$

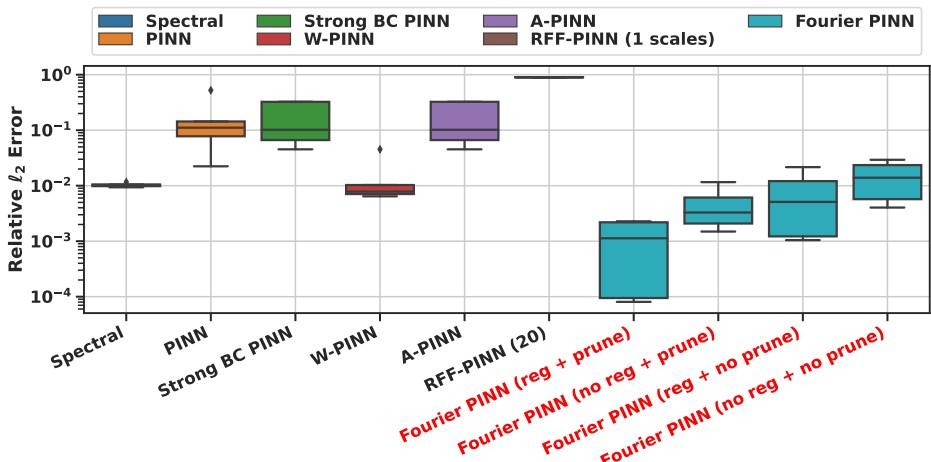

Figure 13: *(1D One-Way Wave) Boxplot of relative $\ell_2$ errors for Fourier PINNs and top-performing baseline methods.*

with a true solution of,

$$u(x,t) = \sin(\pi x)\cos(10\pi t) + \sin(2\pi x)\cos(20\pi t). \tag{43}$$

This problem introduces a time-dependent component, testing the ability of models to handle first-time derivatives within a high-frequency context. We conducted an ablation study to assess the roles of basis pruning and regularization by evaluating Fourier PINNs without these mechanisms. This study highlights the distinct contributions of basis pruning and regularization to model accuracy and computational efficiency. Table 6 in the Appendix lists all hyperparameters and experimental details.

Figure 13 shows the relative $\ell_2$ error distributions for Fourier PINNs and top-performing baseline models across all manufactured solutions. Detailed results for all models are provided in Figure 19 in the Appendix. As expected, Fourier PINNs with both basis regularization and pruning (denoted as "reg + prune") achieve the best results, while the configuration using neither (denoted as "no reg + no prune") yields the worst performance.

**Fourier-PINN Without Basis Pruning.** In configurations without basis pruning, all frequency candidates up to the maximum frequency $K$ were allowed to contribute to the solution. Two variations were tested: with basis regularization ("reg + no prune") and without regularization ("no reg + no prune"). Both configurations resulted in worse relative $\ell_2$ errors than the pruned counterparts, although the regularized version performed slightly better than the unregularized one.

**Fourier-PINN Without Regularization.** To further evaluate the impact of regularization, we tested configurations without the regularization term. Two setups were compared: with basis pruning ("no reg + prune") and without pruning ("no reg + no prune"). The configuration without regularization but with basis pruning ("no reg + prune") achieved better relative $\ell_2$ errors than the setup using regularization but no basis pruning ("reg + no prune"). This result suggests that basis pruning has a slightly more significant influence on solution quality than regularization.

## 6.6 1D Non-Steady-State Allen-Cahn Problem

The time-dependent Allen-Cahn equation is a well-established benchmark for testing the limitations of PINN models, particularly due to its stiffness, sharp transitions, and nonlinear dynamics (Wight & Zhao, 2020; Wang et al., 2024b). For simplicity, we focus on a one-dimensional case with periodic boundary conditions

over $t \in [0, 1]$ and $x \in [-1, 1]$:

$$u_t - 0.0001 u_{xx} + 5u^3 - 5u = 0,$$
$$u(0, x) = x^2 \cos(\pi x),$$
$$u(t, -1) = u(t, 1), \ u_x(t, -1) = u_x(t, 1).$$

This problem is particularly challenging because the Allen-Cahn PDE introduces complex, multi-scale dynamics driven by a combination of stiff reaction terms and spatial diffusion. Unlike problems with harmonic solutions, the nonlinear terms generate sharp transitions and localized structures in both space and time, testing the model's adaptability to varying frequency components and its robustness to sharp gradients. Table 8 in the Appendix outlines the hyperparameter and experimental settings used in the example. In this experiment, we randomly sampled 8192 collocation points within the domain at each iteration. Figure 14 compares the convergence behavior of Fourier PINNs with baseline models (standard PINN and RFF-PINN).

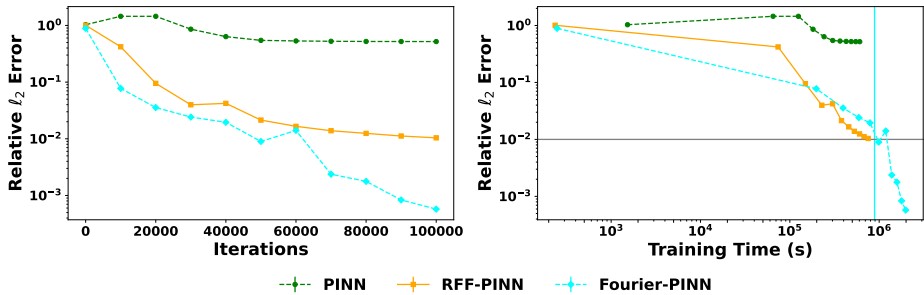

Figure 14: *Convergence behavior of different models on the 1D non-steady-state Allen-Cahn equation. Fourier PINNs achieve superior accuracy and convergence, effectively capturing sharp transitions and localized structures, while baseline methods falter.*

**Results.** As shown in Figure 14, the Fourier PINN significantly outperforms both the standard PINN and RFF-PINN. The relative $\ell_2$ error decreases steadily for Fourier PINNs, reaching values below $10^{-3}$ after sufficient training iterations, while the baseline models plateau at higher error levels. Specifically, the standard PINN struggles with stiffness, failing to reduce the error below $10^{-1}$ even after extended training. The RFF-PINN shows some improvement over the standard PINN but is highly sensitive to scale settings, resulting in slower convergence and larger errors compared to Fourier PINNs. Fourier PINNs leverage the augmented Fourier layer structure to efficiently represent both smooth and sharp features, as evidenced by their rapid convergence and low final error. This experiment underscores the potential of Fourier PINNs to handle challenging nonlinear PDEs with sharp transitions, where conventional PINN architectures struggle.

## 6.7 Discussion

The 1D and 2D problems results reveal that standard PINN, W-PINN, and A-PINN consistently fail to achieve reasonable solutions, with relative $\ell_2$ errors remaining large ($\sim 1.0$) across all cases. This indicates that these methods struggle to capture high-frequency signals effectively. While RFF-PINNs achieve lower relative errors under specific scale configurations, they exhibit significant sensitivity to the choice of the number and range of scales. For most configurations (60–70% of the 20 tested settings listed in Table 2), RFF-PINNs failed with large solution errors. Notably, even in the single-scale solution case (Figure 15a), successful configurations often involved multi-scale settings (e.g., (1, 20, 194)), whereas all single-scale settings failed. These findings highlight the lack of a clear relationship between scale configurations and solution accuracy, posing a significant challenge for RFF-PINNs. In contrast, Fourier PINNs consistently delivered accurate results, with relative $\ell_2$ errors ranging from $10^{-3}$ to $10^{-4}$ across almost all test cases. A key advantage of Fourier PINNs is their robustness to the choice of the frequency range $K$. Provided $K$ is sufficiently large, the method reliably selects the target frequencies, achieving comparable solution accuracy across cases. It is important to note that Fourier PINNs do not employ advanced loss re-weighting schemes (e.g., NTK re-weighting used in RFF-PINNs or mini-max updates) in this study, as the focus was on evaluating

the core algorithm. However, such enhancements could be seamlessly integrated into the Fourier PINN framework to further improve accuracy. Additionally, Fourier PINNs outperform baseline methods in higher-dimensional problems, demonstrating scalability and the ability to handle high-frequency and multi-scale solutions. However, their performance on (some) nonlinear, higher-dimensional PDEs, such as the 2D Allen-Cahn example in Equation 41, highlights areas for future work. Specifically, improving training efficiency and refining the network architecture to better address the challenges posed by nonlinear dynamics and stiffness could further enhance the applicability of Fourier PINNs.

We finally visualize the solution of each method in solving a 2D Poisson equation (see Figure 17b) to visualize the point-wise errors for each method within the two-dimensional domain. The ground-truth solution is given in Figure 20, and the solution obtained by each method is shown in Figure 21. While the spectral method also obtains good results, the point-wise error biases much on the regions close to the boundary, and the relative $\ell_2$ error is two orders of magnitude larger than Fourier PINN's. RFF-PINN (with the best number and scale choice) can roughly capture the solution structure, but the accuracy is far worse than that of Fourier PINNs and the spectral method. For example, in Figure 17b, RFF-PINN nearly failed with every choice of the number and scale set. We can see that Fourier PINNs best recover the original solution.

## 7 Future Work: Improving Scalability with Tensor Decomposition Techniques

One limitation of our current approach is its scalability in higher-dimensional settings. The computational cost grows exponentially with the problem's dimensionality, making it infeasible for very high-dimensional cases. Future work could explore tensor decomposition techniques, such as Tensor-Train (TT) decomposition and Tucker decomposition, to reduce the computational complexity and memory requirements.

Specifically, in high-dimensional spaces, the solution $u(x_1, x_2, \ldots, x_d)$ can be represented as a tensor-product over basis functions $\phi(x_i)$, with the weights forming a high-dimensional tensor $\mathcal{U}$:

$$u(x_1, x_2, \ldots, x_d) \approx \sum_{i_1=1}^{n_1} \sum_{i_2=1}^{n_2} \cdots \sum_{i_d=1}^{n_d} \mathcal{U}_{i_1, i_2, \ldots, i_d} \, \phi_{i_1}(x_1) \, \phi_{i_2}(x_2) \cdots \phi_{i_d}(x_d). \tag{44}$$

Without decomposition, storing and computing with $\mathcal{U}$ requires $\prod_{j=1}^{d} n_j$ parameters, which is impractical in high dimensions.

One promising approach to mitigate this is through Tucker decomposition, where $\mathcal{U}$ is decomposed into a core tensor $\mathcal{C}$ and a set of factor matrices $\{\mathbf{A}_j\}$:

$$\mathcal{U}_{i_1, i_2, \ldots, i_d} \approx \sum_{j_1=1}^{r_1} \sum_{j_2=1}^{r_2} \cdots \sum_{j_d=1}^{r_d} \mathcal{C}_{j_1, j_2, \ldots, j_d} \, \mathbf{A}_1(i_1, j_1) \, \mathbf{A}_2(i_2, j_2) \cdots \mathbf{A}_d(i_d, j_d), \tag{45}$$

where each factor matrix $\mathbf{A}_j \in \mathbb{R}^{n_j \times r_j}$ maps the original tensor to a lower-dimensional representation. This decomposition reduces the number of parameters to $\sum_{j=1}^{d} n_j r_j + \prod_{j=1}^{d} r_j$, allowing scalability as it grows linearly with $d$ under fixed ranks $r_j$. Another promising approach is Tensor-Train (TT) decomposition, where $\mathcal{U}$ is expressed as a chain of lower-order tensors, known as TT-cores:

$$\mathcal{U}_{i_1, i_2, \ldots, i_d} = \sum_{\alpha_1=1}^{r_1} \sum_{\alpha_2=1}^{r_2} \cdots \sum_{\alpha_{d-1}=1}^{r_{d-1}} \mathbf{G}_1(i_1, \alpha_1) \, \mathbf{G}_2(\alpha_1, i_2, \alpha_2) \cdots \mathbf{G}_d(\alpha_{d-1}, i_d), \tag{46}$$

where $\mathbf{G}_1 \in \mathbb{R}^{n_1 \times r_1}$, $\mathbf{G}_d \in \mathbb{R}^{r_{d-1} \times n_d}$, and intermediate cores $\mathbf{G}_j \in \mathbb{R}^{r_{j-1} \times n_j \times r_j}$ for $j = 2, \ldots, d-1$. The TT-ranks $r_j$ control the complexity of the representation, enabling a linear scaling of parameters with $d$ for fixed ranks.

## 8 Conclusion

We have presented Fourier PINNs, a novel extension to capture high-frequency and multi-scale solution information. Our adaptive basis learning and selection algorithm can automatically identify target frequencies

and truncate useless ones without tuning the number and scales. We assume the spacing (granularity) of the candidate frequencies is fine-grained enough, which can be overly optimistic. In the future, we plan to develop adaptive spacing approaches to capture this granularity. Applying tensor decomposition techniques, such as Tucker and Tensor-Train, to the weight tensor $\mathcal{U}$ over basis functions could provide a viable path for extending our approach to higher dimensions. By reducing the effective dimensionality of the parameter space, these methods offer a promising direction to overcome scalability challenges and maintain the advantages of meshless approaches in high-dimensional spaces.

**Acknowledgments**

We acknowledge the use of large language models (LLMs) for manuscript editing. This work has been supported by MURI AFOSR grant FA9550-20-1-0358, NSF CAREER Award IIS-2046295, and NSF OAC-2311685 and the Army Research Office (ARO).

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

## 9 Additional Experimental Details and Results

Table 3: *Hyper-parameter configurations for 1D Poisson and 1D Steady-State Allen-Cahn frequency sweep experiments.*

| Parameter | Value |
| --- | --- |
| **Architecture** | |
| Number of layers | 2,4 |
| Number of channels | 100 |
| Activation | Tanh |
| **Learning rate schedule** | |
| Initial Adam learning rate | $10^{-3}$ |
| Initial LBFGS learning rate | $10^{-2}$ |
| Decay rate | 0.9 |
| Decay steps | every 1000 |
| Decay type | exponential |
| **Training** | |
| Training steps | $100K$ |
| Collocation points | $10K$ |
| Testing points | $20K$ |

Table 4: *Hyper-parameter configurations for 1D Poisson analysis experiments.*

| Parameter | Value |
|---|---|
| **Architecture** | |
| Number of layers | 4 |
| Number of channels | 100 |
| Activation | `Tanh` |
| **Learning rate schedule** | |
| Initial Adam learning rate | $10^{-3}$ |
| Initial LBFGS learning rate | $10^{-2}$ |
| Decay rate | 0.9 |
| Decay steps | every 1000 |
| Decay type | exponential |
| **Training** | |
| Training steps | $1 \times 10^5$ |
| Collocation points | $10K$ |

Table 5: *Hyper-parameter configurations for 1D Poisson and 1D Steady-State Allen-Cahn experiments.*

| Parameter | Value |
|---|---|
| **Architecture** | |
| Number of layers | 2 |
| Number of channels | 100 |
| Activation | `Tanh` |
| **Learning rate schedule** | |
| Initial Adam learning rate | $10^{-3}$ |
| Initial LBFGS learning rate | $10^{-2}$ |
| Decay rate | 0.9 |
| Decay steps | every 1000 |
| Decay type | exponential |
| **Training** | |
| Training steps | $4 \times 10^4$ |
| Collocation points | $10K$ |
| Testing points | $20K$ |

Table 6: *Hyper-parameter configurations for 1D Wave experiments.*

| Parameter | Value |
| --- | --- |
| **Architecture** | |
| Number of layers | 3 |
| Number of channels | 128 |
| Activation | `Tanh` |
| **Learning rate schedule** | |
| Initial Adam learning rate | $10^{-3}$ |
| Initial LBFGS learning rate | $10^{-2}$ |
| Decay rate | 0.9 |
| Decay steps | every 1000 |
| Decay type | exponential |
| **Training** | |
| Training steps | $4 \times 10^4$ |
| Collocation points | $5K$ |
| Initial condition points | 256 |
| Boundary points | 200 |
| Testing points | $100K$ |

Table 7: *Hyper-parameter configurations for 2D Poisson and 2D Steady-State Allen-Cahn experiments.*

| Parameter | Value |
| --- | --- |
| **Architecture** | |
| Number of layers | 3 |
| Number of channels | 100 |
| Activation | `Tanh` |
| **Learning rate schedule** | |
| Initial Adam learning rate | $10^{-3}$ |
| Initial LBFGS learning rate | $10^{-2}$ |
| Decay rate | 0.9 |
| Decay steps | every 1000 |
| Decay type | exponential |
| **Training** | |
| Training steps | $4 \times 10^4$ |
| Collocation points | $12K$ |
| Boundary points | 400 |
| Testing points | $100K$ |

Table 8: *Hyper-parameter configurations for 1D (non-Steady-State) Allen-Cahn experiments.*

| Parameter | Value |
|---|---|
| **Architecture** | |
| Number of layers | 4 |
| Number of channels | 128 |
| Activation | Tanh |
| **Learning rate schedule** | |
| Initial Adam learning rate | $10^{-3}$ |
| Initial LBFGS learning rate | $10^{-2}$ |
| Decay rate | 0.9 |
| Decay steps | every 1000 |
| Decay type | exponential |
| **Training** | |
| Training steps | $10^5$ |
| Batchsize | 8192 |
| Boundary points | 200 |
| Initial condition points | 256 |
| Testing points | $100K$ |
| **Weighting** | |
| Weighting scheme | Gradient Norm Wang et al. (2022) |
| Causal training | True, Wang et al. (2024b) |
| Causal tolerance | 1.0 |
| Number of chunks | 32 |

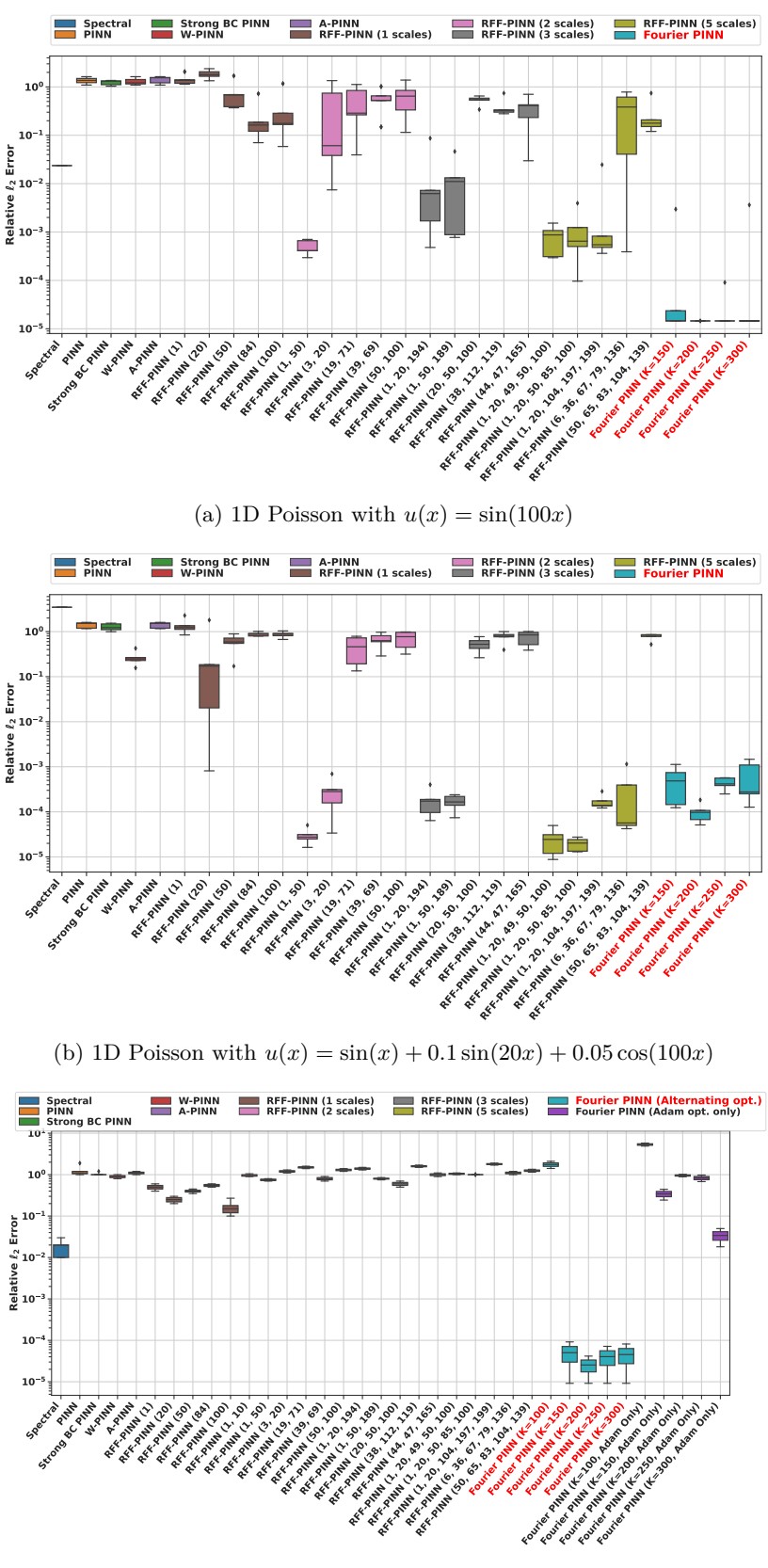

(a) 1D Poisson with $u(x) = \sin(100x)$

(b) 1D Poisson with $u(x) = \sin(x) + 0.1\sin(20x) + 0.05\cos(100x)$

(c) 1D Poisson with $u(x) = \sin(6x)\cos(100x)$

Figure 15: Combined figures for different 1D Poisson configurations.

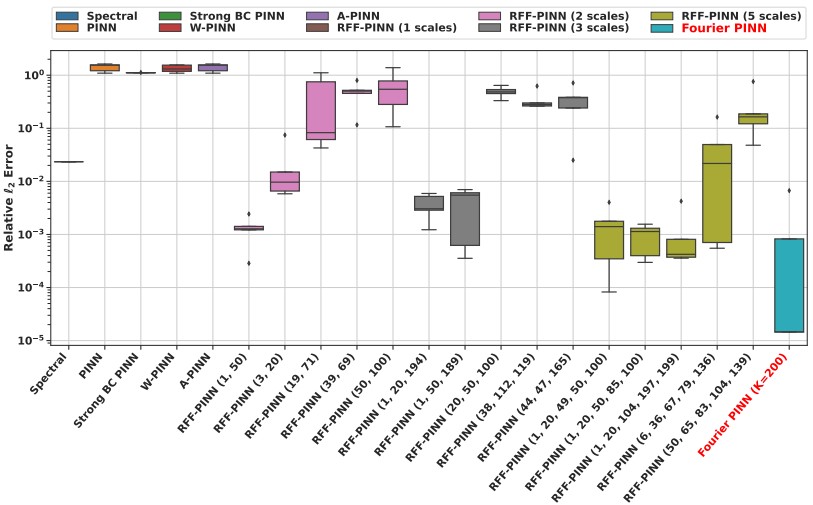

(a) 1D Allen-Cahn equation with true solution $u(x) = \sin(100x)$

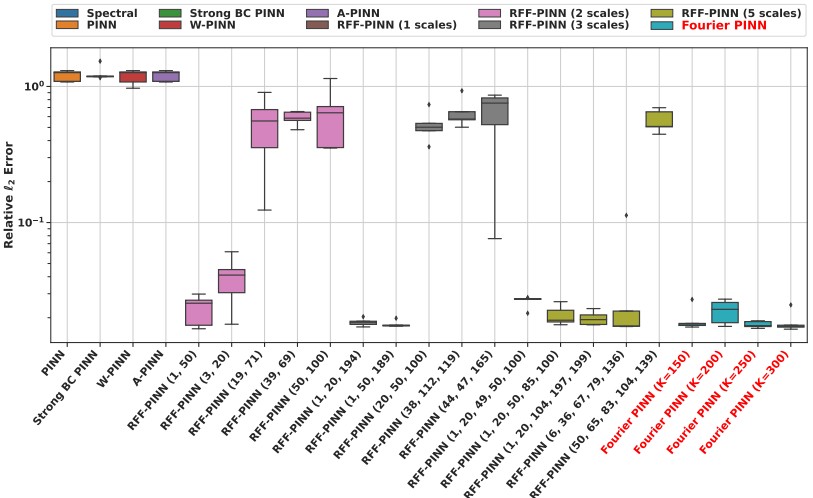

(b) 1D Allen-Cahn with true multi-scale solution $u(x) = \sin(x) + 0.1\sin(20x) + 0.05\cos(100x)$

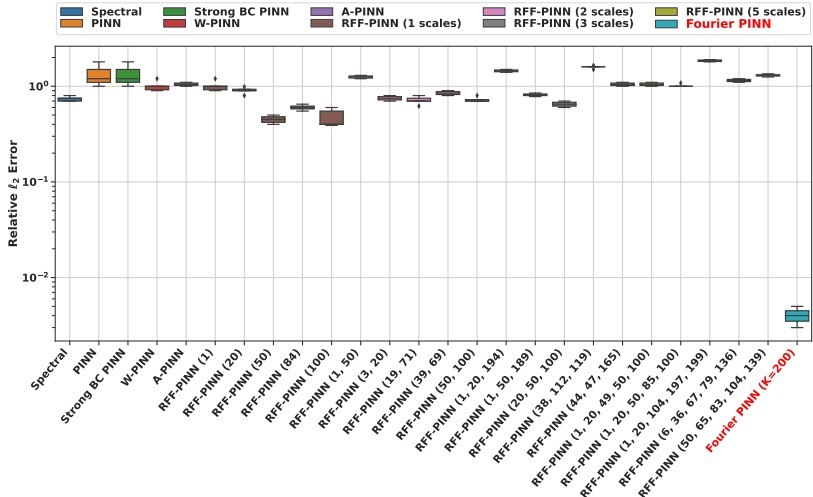

(c) 1D Allen-Cahn equation with true solution $u(x) = \sin(6x)\cos(100x)$

Figure 16: Combined figures for different 1D Allen-Cahn configurations.

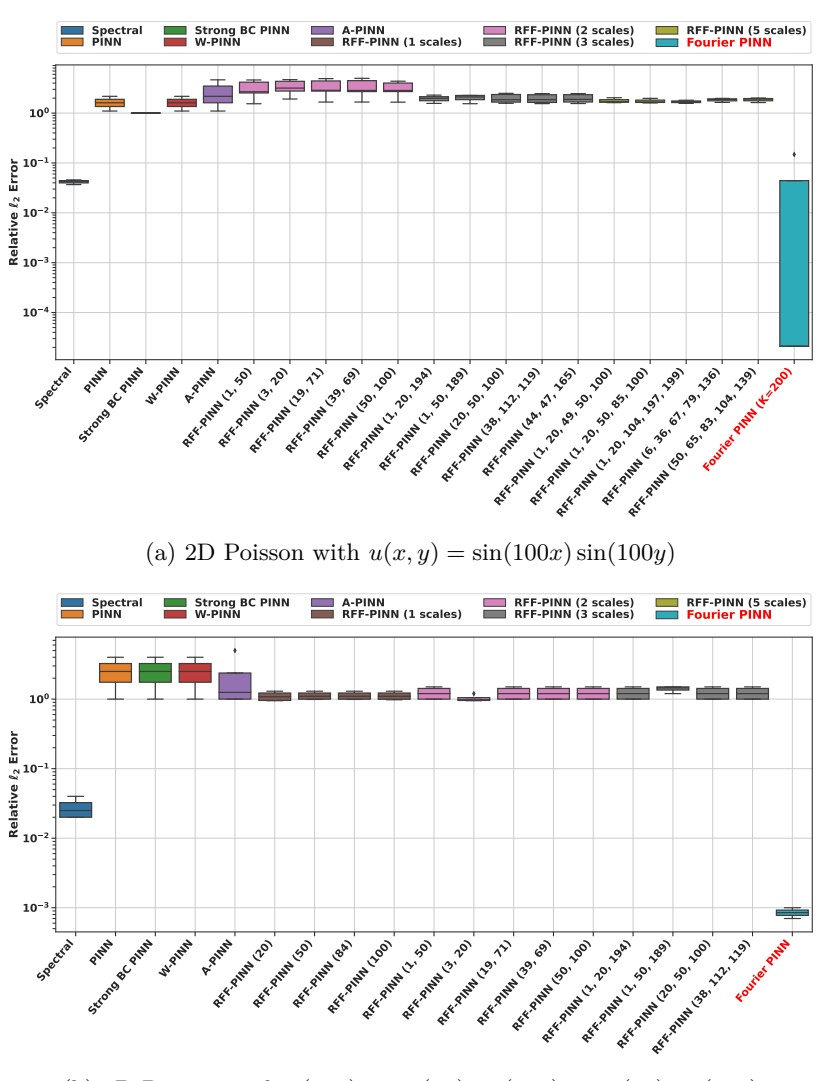

(a) 2D Poisson with $u(x, y) = \sin(100x)\sin(100y)$

(b) 2D Poisson with $u(x, y) = \sin(6x)\cos(20x) + \sin(6y)\cos(20y)$

Figure 17: Combined figures for different 2D Poisson configurations.

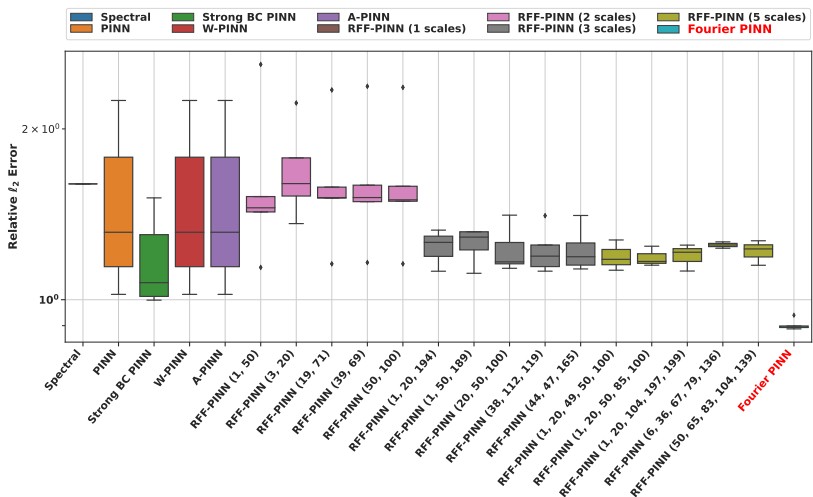

Figure 18: 2D Allen-Cahn with $u(x, y) = (\sin(x) + 0.1\sin(20x) + \cos(100x)) \cdot (\sin(y) + 0.1\sin(20y) + \cos(100y))$

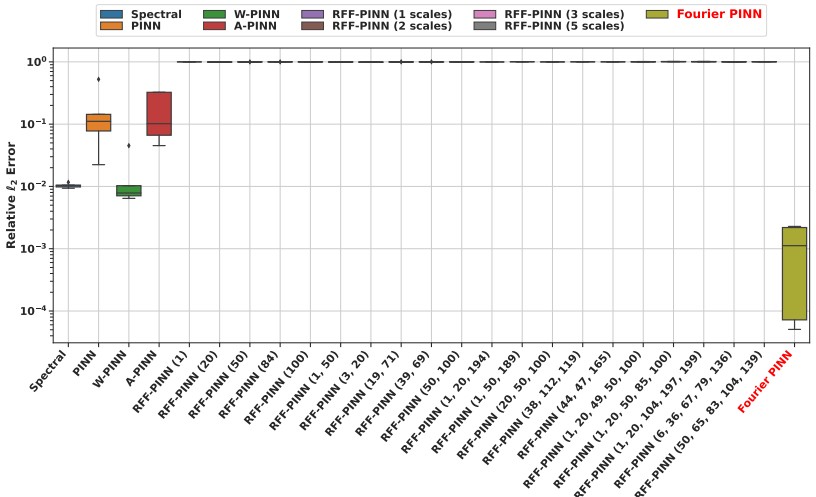

Figure 19: 1D One-Way Wave equation with true solution $u(x, t) = \sin(\pi x)\cos(10\pi t) + \sin(2\pi x)\cos(20\pi t)$

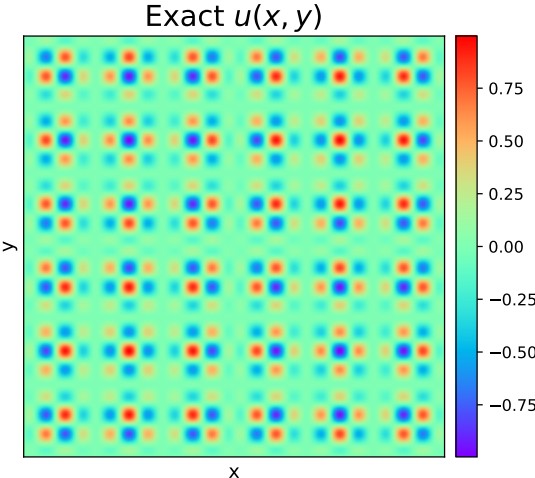

Figure 20: 2D Poisson equation with true solution $u(x, y) = \sin(6x)\cos(20x) + \sin(6y)\cos(20y)$

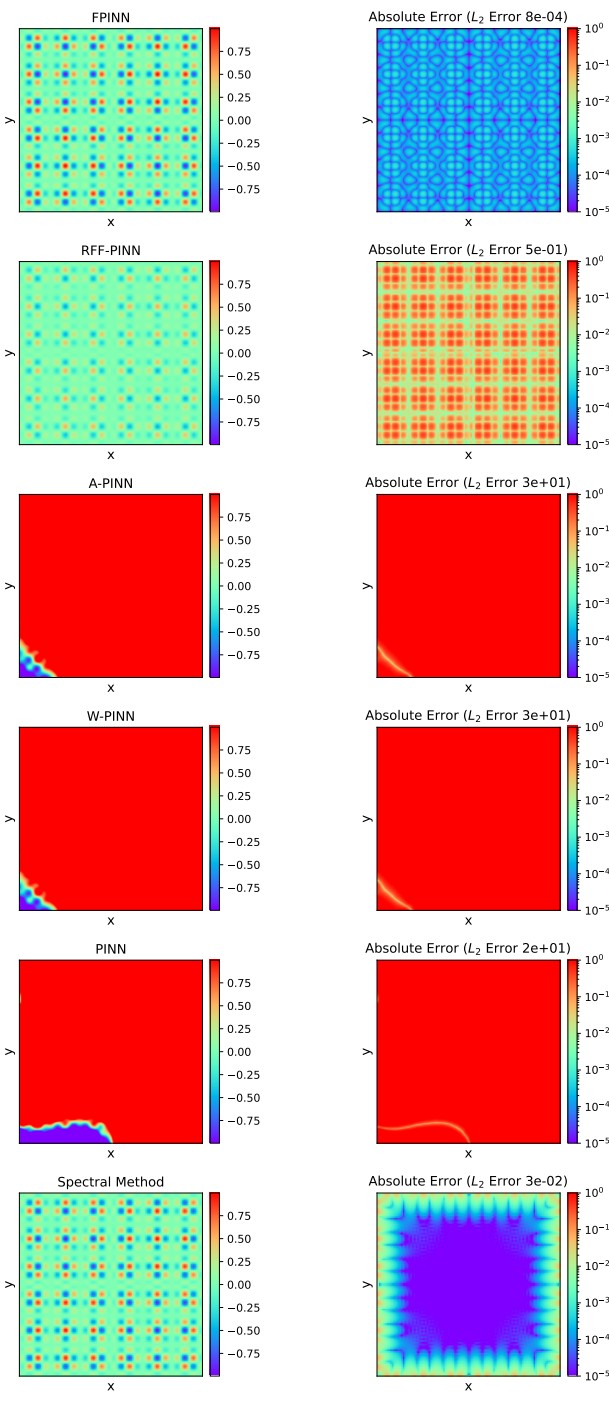

Figure 21: Predicted solutions (left) and absolute errors (right) of each method on the 2D Poisson equation with true solution $u(x, y) = \sin(6x)\cos(20x) + \sin(6y)\cos(20y)$

