# OpenReview forum: "Fourier PINNs: From Strong Boundary Conditions to Adaptive Fourier Bases"
_TMLR — Accepted by TMLR_

### Review · Reviewer_HXi7 · 2024-09-05

**Summary Of Contributions:**

The authors introduce two distinct approaches for solving PDEs using PINNs. The first method enforces boundary conditions by incorporating simple distance functions, demonstrated in 1D domains with simple polynomial combinations and tunable exponential terms to adjust the influence on boundary conditions relative to the internal domain.
The second approach, termed Fourier PINNs, combines a Fourier series expansion with a neural network by superposition. The coefficients for both the Fourier basis and the neural network are trained concurrently.
While the initial motivation of the paper appears to focus on handling boundary condition challenges, the work ultimately shifts towards approximating high-frequency solutions of PDEs. The central challenge remains somewhat ambiguous, as the narrative oscillates between mitigating high-frequency components as noise and focusing on high-frequency solution examples, such as sin(100x) and more complex functions.
The treatment of general PDE solutions is limited, as the examples predominantly involve smooth trigonometric functions. The scope of examples is also somewhat narrow, with multiple cases feeling repetitive and not contributing additional insights.

**Audience:**

Yes

**Broader Impact Concerns:**

There are no broader ethical concerns or societal implications associated with this work, as it primarily focuses on technical methods for solving partial differential equations using machine learning techniques. The paper does not involve sensitive data, user interaction, or applications that could raise ethical concerns. Therefore, no broader impact statement is required in this context.

**Claims And Evidence:**

No

**Requested Changes:**

Strengthen the Experimental Section:

•	Expand the experimental section by including a broader range of test cases, particularly PDEs with non-periodic solutions.

•	Provide comparative results against traditional numerical methods such as finite difference (FD), finite element methods (FEM), and spectral methods. If a direct comparison is not feasible, at least acknowledge the efficiency and accuracy of these well-established methods, especially in low-dimensional settings. It is important that the paper does not overstate the computational efficiency of the proposed methods, as FD and FEM often outperform PINNs in terms of computational cost and accuracy, particularly for simpler problems.

Acknowledge Limitations and Alternative PINNs Approaches:

•The paper should explicitly acknowledge the limitations of the proposed methods, particularly in terms of scalability, computational cost, and performance relative to traditional methods like FD and FEM. It is important to clarify that while meshless methods like PINNs hold promise, they are not currently competitive with FD, FEM, or spectral methods in many scenarios, especially in low dimensions or simple geometries.

•Additionally, the author should acknowledge the existence of more advanced and computationally efficient PINNs approaches, such as those based on random projections, random features, randomized networks, and extreme learning machines.

Clarify Methodology and Assumptions:

•Clearly explain how the adaptive distance function method could be extended to higher dimensions and different types of boundary conditions.

•Provide additional details on how the Fourier PINNs method handles nonlinearity, particularly in the case of the Allen-Cahn equation.

•Correct the error regarding the L2 regularization claim.

Improve Reproducibility.

Clarify and Correct Minor Issues

Revise and simplify Figures for better clarity.

**Strengths And Weaknesses:**

Strengths:

•The paper provides a comparative study of different methods and implementations of PINNs, evaluating their performance against numerous baseline approaches. This adds clarity to the results.

•The use of an adaptive distance function could be considered a novel idea, though its originality may be debatable. It bears similarity to concepts from numerical analysis and might represent a minor contribution.

•Both the strong boundary condition PINNs and Fourier PINNs achieve competitive error rates in the experiments compared to vanilla PINNs. While offering small novel contributions, this work seems more like an academic exploration focused on high-frequency and multifrequency PDE solutions.

Weaknesses: Given the two methods proposed, I will outline the weaknesses of each method separately, followed by those that apply to both.

Weaknesses of Method 1 (Strong Boundary Conditions):

•The adaptation of the basis is the primary contribution of this method, but this may be a minor and insufficient advancement.

•The change in steepness, while potentially valuable, is symmetric and cannot independently adapt to different boundary conditions, which may require different steepness levels.

•This approach is demonstrated only for 1D problems, and it remains unclear how to extend the proposed distance function to higher dimensions. While generating such functions in higher dimensions is inherently challenging, this limitation restricts the method's broader applicability.

•The method's extension to more complex boundary conditions, such as Neumann, Robin, or nonlinear conditions, is not addressed. It is evident that a soft constraint or a more formal approach, such as using Lagrange multipliers, might be necessary.

•The Fourier spectrum analysis, in section 4, is limited to a single solution, sin(15x). Real PDE solutions often have continuous spectra rather than discrete points in Fourier space, so the applicability of this analysis to more complex situations remains unclear.

•The trial function incorporates the boundary function g (i.e., u=g+phi⋅u_n), but the analysis is restricted to the homogeneous case where g=0. It is unclear how this method would extend to nonzero g, especially in higher dimensions where the Fourier transform of g could complicate the analysis.

•Figure 2 results lack clarity. The relative l2 error is not well-defined in the text and should be more explicitly described. Additionally, the x-axis values do not match the text's referenced values, though this is a minor issue. The relative l2 error appears larger than expected and is not compared to higher-order methods. Crucial details, such as the training method, learning rate, and number of epochs, are not sufficiently explained. Furthermore, the large number of collocation points (10K) raises concerns about computational cost. A comparison with classical techniques like finite difference (FD) or finite element methods (FEM) is missing.

•The "threshold" frequency, at which the method's performance degrades, is not clearly presented in the figures. The method seems to deteriorate gradually but remains better than simple PINNs, yet further clarity is needed.

•The method's error does not improve beyond 10^−5, especially for higher k, which accuracy is quite limited. It would be helpful to explore whether larger networks could achieve better convergence. The relatively high error rates for high k suggest that comparing this method to FD, FEM, or spectral methods under the same conditions would provide more meaningful insights. The lack of such comparisons leaves the impression that the method might not be efficient, even for simple 1D problems, let alone in higher dimensions or more complex geometries.

Weaknesses of Method 2 (Fourier PINNs):

•The formulation in Eq. 21, which superimposes two bases of functions, differs from the more common approach of using Fourier features as additional hidden layers to enhance solution features while retaining the spatial variable x. What is proposed seems more like a correction approach. The simultaneous optimization, despite alternating between the two bases, might not allow sufficient time for convergence. It is unclear which term will dominate, possibly depending on initialization. Given the universal approximation property of PINNs, both bases could potentially identify the solution, and the issue with the slow spectral decay of PINNs may still persist. A spectral analysis here would have been valuable.

•The numerical examples mostly involve trigonometric functions, making the use of Fourier bases appear advantageous, even artificial. Testing the method on non-periodic functions, such as sigmoids or Gaussians, would provide a more balanced evaluation.

•A detailed computational cost analysis should be included. How does the training cost increase when Fourier features are introduced? Are they faster to converge, and how computationally demanding is each epoch? Reporting execution times and the hardware used (CPU, GPU, or clusters) would provide valuable context.

•The approach becomes computationally infeasible in higher dimensions, as stated in eq 22, which is a significant drawback. Meshless methods are expected to be advantageous in high dimensions, yet this method appears to struggle with scalability.

•The introduction of L2 regularization in Eq. 23 is problematic. The claim that L2 regularization promotes sparsity is incorrect. Sparsity is typically achieved with L1 regularization. There is no evidence that L2 regularization yields sparse solutions, making the claim unfounded.

•In Eq. 24, the focus is on linear operators, but the paper also addresses nonlinear PDEs, such as the Allen-Cahn equation. The treatment of nonlinearity in Eq. 28 seems inadequate, as it overlooks the nonlinear term, effectively solving a different PDE. This raises concerns about convergence, and the method for handling nonlinearity is not clearly described.

Weaknesses of Both Methods:

•The comparisons are limited to other PINN-based methods. Including a comparison with traditional methods like FD, FEM, or spectral methods would provide a more comprehensive evaluation. The claim that these methods are computationally efficient alternatives to traditional numerical methods is overstated and should be removed. While promising, they are not yet efficient in practice.

•The comparison seems overly focused on RFF-PINNs, but this is not the original Random Fourier Features (RFF) approach as described by Rahimi and Recht (2007-2008). The current implementation only partially incorporates randomness, making the figures difficult to interpret. Reducing the number of RFF-PINN comparisons or presenting the results in a table would help.

•The authors appear unaware of alternative PINN approaches based on random features, such as Random Vector Functional Link networks (RVFLN), Extreme learning machines (ELM), Random Projection neural networks (RPNN), Randomized Netwokrs, or Reservoir Computing, which have shown greater computational efficiency. Key references, such as the works of the group of (a) Constantinos Siettos, (b) Suchuan Dong, (c) Felix Dietrich (among others), especially for the case of PDEs, and recent advances like RandOnet for learning operators in few seconds with high accuracy, should be considered for a more thorough discussion. Indeed such method, should be the real counterpart PINN approaches where the author have to compare their method. This is the state-of-the-art. Not vanilla PINNs. At least in 1-2 dimension problems.

•The description of the “spectral” method, i.e. named spectral in figures, in the paper lacks clarity. The algorithm used to compute the coefficients is not well-explained. For nonlinear problems, optimization may not be the best approach; Newton's method or straightforward linearization could be more appropriate. The use of Dirichlet boundary conditions with a Fourier basis is also unclear, and a Chebyshev spectral approach might be more suitable. The poor results with the spectral method could mislead naïve readers about its effectiveness for simple PDEs.

•The impact of varying the number of collocation points, neurons, and hidden layers should be investigated. The iterative nature of PINN training means convergence behavior needs to be reported in detail. Different initializations often yield varying results, and while the shaded regions in Figure 2 might address this variability, the context needs to be clarified.
•Finally, the relative l2 error, though mentioned, does not specify the mesh of points on which it is computed. This crucial detail should be clearly communicated.

Clarity, Quality, Novelty, and Reproducibility:

The paper is generally well written and presents its ideas with clarity. However, there are areas where more information is needed to fully assess the significance, novelty, and reproducibility of the proposed methods.

Clarity:

The manuscript is clearly structured, and the authors explain their ideas effectively. However, the limited experimental section hinders a deeper understanding of the real impact of the proposed approaches. The clarity of the presentation would benefit greatly from more detailed and varied experimental results. Specifically, exploring a wider range of PDEs beyond simple trigonometric functions would provide stronger support for the claims made in the paper.

Quality:

The experimental evaluation is rather weak, making it difficult to assess the practical value of the methods. The lack of comparisons with traditional numerical methods, such as finite difference (FD), finite element methods (FEM), or spectral methods, leaves a gap in understanding how these approaches perform relative to well-established techniques. The omission of a detailed computational cost analysis further limits the ability to gauge the efficiency of the proposed methods. The experiments provided are confined to basic examples, and the absence of a broader evaluation against diverse PDEs reduces the generalizability of the findings.

Novelty:

The paper introduces some ideas, particularly the adaptive distance functions and the Fourier PINNs. However, the novelty of these contributions seems limited. The adaptive distance function could be seen as a small extension of existing numerical techniques, and its impact appears minor. The Fourier PINNs approach, while intriguing, largely focuses on trigonometric functions, which naturally fit into a Fourier framework. Testing the method on non-periodic solutions would help better establish its general applicability. Additionally, the handling of nonlinearity in PDEs, especially with the Allen-Cahn equation, raises concerns about whether the proposed method is truly advancing the state of the art. Overall, the contributions, though potentially useful, might be perceived as incremental rather than groundbreaking.

Reproducibility:

While the paper is well written, it lacks the necessary details to ensure that the methods can be easily replicated. Key information such as training methods, learning rates, number of epochs, hardware specifications (e.g., CPU, GPU), and the precise number of collocation points should be explicitly stated. Additionally, the paper would greatly benefit from releasing the code alongside the manuscript.

Minor Points:

•The domain of PDEs is stated as [0, 2pi], but the left boundary conditions appear to be applied at x=1 in equations 12 and 14. This seems inconsistent with the stated domain and should be clarified.

---

> ### Author Response · Authors · 2024-10-23
> **Response to Reviewer HXi7 (Part 1)**
>
> We appreciate the positive feedback regarding the background and introduction to PINNs along with the theoretical foundation provided by the Fourier analysis, and the adaptive basis selection.
>
> **Regarding the Weaknesses of Method 1**
>
> > The adaptation of the basis is the primary contribution of this method, but this may be a minor and insufficient advancement.
>
>  While the adaptive distance function we propose in this work might appear incremental, our main aim was to demonstrate a more scalable and flexible strong boundary condition architecture design that provides a clear improvement over the traditional PINNs and existing (non-adaptive) distance functions for exactly imposing Dirichlet boundary conditions. We acknowledge that the novelty may seem limited, and **we have emphasized this in the revised manuscript (see Section 2.2, paragraph 6).**
>
> > The change in steepness, while potentially valuable, is symmetric and cannot independently adapt to different boundary conditions, which may require different steepness levels.
>
> The current presentation of the adaptive steepness parameters is certainly symmetric, however, a simple extension is to incorporate different parameters for each boundary edge, thus allowing for different steepness levels and removing the symmetric requirement. In the revised manuscript, **we have included a discussion of how asymmetric functions could be incorporated into the framework to allow for different steepness levels at different boundary points (see Section 2.2, paragraph 6).**
>
> > This approach is demonstrated only for 1D problems, and it remains unclear how to extend the proposed distance function to higher dimensions. While generating such functions in higher dimensions is inherently challenging, this limitation restricts the method's broader applicability. The method's extension to more complex boundary conditions, such as Neumann, Robin, or nonlinear conditions, is not addressed. It is evident that a soft constraint or a more formal approach, such as using Lagrange multipliers, might be necessary.
>
> We recognize the need for more discussion regarding extensions to higher dimensions and **have included discussions** about the potential strategies for generalizing the distance function to higher-dimensional domains. Additionally, we **have included a discussion describing** potential future directions using methods to handle irregular domains based on Fourier extension methods or by leveraging custom network layers that map to rectilinear domains, similar to those used in the GOFNO network. We agree that this is an important aspect that requires further development regarding the extension to complex boundary conditions, such as Neumann, Robin, or nonlinear conditions. We **have discussed** related and potential methods for handling these scenarios. For example, Neumann boundary conditions can be incorporated by constructing distance functions with non-zero slope at the boundaries, though this would require a careful and non-trivial construction **(see Section 2.2, paragraph 7)**.
>
> > The Fourier spectrum analysis, in section 4, is limited to a single solution, sin(15x). Real PDE solutions often have continuous spectra rather than discrete points in Fourier space.
>
> Thank you for raising this concern. We **have included** a similar Fourier spectrum analysis for solutions with more complex behavior to demonstrate its applicability further. **Specifically, in Section 4.1, we have included similar Fourier analysis for three additional cases including: a combination of sine waves with differing frequencies to explore how multi-scale components are captured, a Gaussian-modulated sine wave to evaluate performance on solutions with continuous spectral content, and a sine wave and polynomial combination to examine behavior under non-homogeneous boundary conditions.**
>
> > The trial function incorporates the boundary function g (i.e., $u=g+phi⋅u_n$), but the analysis is restricted to the homogeneous case where g=0. It is unclear how this method would extend to nonzero g.
>
> We **have extended** our analysis to include non-homogeneous cases where $g \neq 0$. This demonstrates the method's applicability to more realistic boundary conditions and provide insight into how $g$ affects the boundary function and analysis conclusions. We **have included** a discussion on higher dimensions **(see Section 4.1)**.
>
> > Regarding training details and traditional methods.
>
> Thank you for your comment. We **have revised** the figure and clarify the definition of the relative $\ell_2$ error in the text. Additionally, **we have corrected** the discrepancies between the x-axis values and the text and provide more detailed information on the training setup, including the learning rate, number of epochs, and other relevant experimental details. We **have** also added a discussion on computational cost and compare it to traditional methods **(see Section 3)**.

---

> ### Author Response · Authors · 2024-10-23
> **Response to Reviewer HXi7 (Part 2)**
>
> **Regarding the Weaknesses of Method 1 (cont.)**
>
> > The "threshold" frequency, at which the method's performance degrades, is not clearly presented in the figures.
>
> We acknowledge that the term “threshold frequency” may not provide the most precise description of the results. We **have clarified** in the discussion of these particular results that while the method’s performance gradually degrades as the frequency increases, it consistently outperforms standard PINNs and that the adaptive distance function variant of the strong BC PINNs consistently outperforms the polynomial/static variant **(see Section 3)**.
>
> > The method's error does not improve beyond 10^−5, especially for higher k, which accuracy is quite limited. It would be helpful to explore whether larger networks could achieve better convergence. The relatively high error rates for high k suggest that comparing this method to FD, FEM, or spectral methods under the same conditions would provide more meaningful insights.
>
> We acknowledge the limitations in the error rates and **have provided** additional results using larger network architectures to explore whether they improve convergence. **Specifically, we have included additoinal results for networks with four hidden layers and found that slight improvements in error but longer training times.**
>
> Further, the limitation of errors around 10^-5 is somewhat inherent to neural networks in the PINN domain due to challenges related to the network's capacity to fully resolve high-frequency components in the solution. Several studies have highlighted this phenomenon, indicating that while PINNs are highly flexible for solving PDEs, their accuracy tends to plateau around this level due to generalization and training limitations (Mishra & Molinaro, 2020), (De Ryck & Mishra, 2021). Recent work has addressed this issue using strategies like adaptive sampling and advanced regularization techniques (Gao et al.).
>
> Regarding the suggestion to compare the method with traditional techniques, we agree that these comparisons would provide a broader context for evaluating the method’s efficiency. While classical numerical methods often outperform PINNs in terms of accuracy for simple 1D problems, PINNs excel in handling more complex geometries, higher dimensions, or problems with noisy or incomplete data (Cai et al., 2021). The flexibility of neural networks allows them to be more easily extended to such scenarios without requiring the grid refinement or meshing typically needed by FD or FEM methods (Pang et al., 2018). That said, we **have included** comparisons with traditional methods to highlight both the strengths and limitations of PINNs under comparable conditions **(see Section 3)**.
>
> We appreciate the reviewer’s insights and **have expanded** on these aspects in the revised manuscript.
>
>
> **Regarding the Weaknesses of Method 2 (Fourier PINNs)**
>
> > The formulation in Eq. 21, which superimposes two bases of functions, differs from the more common approach of using Fourier features as additional hidden layers to enhance solution features while retaining the spatial variable x. The simultaneous optimization, despite alternating between the two bases, might not allow sufficient time for convergence. Given the universal approximation property of PINNs, both bases could potentially identify the solution, and the issue with the slow spectral decay of PINNs may still persist. A spectral analysis here would have been valuable.
>
> We appreciate the reviewer’s insight on this point. We **have added** a spectral analysis to provide more insight into the convergence properties of the superimposed bases **(see Figures 7c/d and 8c/d).**
>
> In addition to the spectral analysis, we **have included** an ablation study that investigates the different **optimization strategies for the basis coefficients. Specifically, we have provided results showing the training convergence behavior of Fourier PINNs using the proposed alternating optimization strategy compared to standard optimization (that doesn't include the least-squares coefficient optimization) in Figure 9, and find that the least-squares step is crucial for coefficient estimation.**
>
> We also acknowledge that while both the neural network and the Fourier basis have universal approximation capabilities, it is well-known that NNs tend to converge more slowly to higher-frequency components due to “spectral bias.” Several works have shown examples where neural networks prioritize learning low-frequency components of the solution before refining (or simply failing to find) the higher-frequencies. Because of this, we believe that the Fourier basis, which is inherently better suited for capturing high-frequency modes, will act as a complementary feature, allowing the network to identify these components more efficiently—which we confirm through our numerical studies.

---

> ### Author Response · Authors · 2024-10-23
> **Response to Reviewer HXi7 (Part 3)**
>
> **Regarding the Weaknesses of Method 2 (cont)**
>
> > The numerical examples mostly involve trigonometric functions, making the use of Fourier bases appear advantageous, even artificial. Testing the method on non-periodic functions, such as sigmoids or Gaussians, would provide a more balanced evaluation.
>
> We agree that the focus on trigonometric functions limits the general applicability of our results. In the revised manuscript, we **have included** experiments on non-trigonometric functions to demonstrate the versatility of the Fourier PINNs approach **(see Section 6.6)**.
>
> > A detailed computational cost analysis should be included. How does the training cost increase when Fourier features are introduced? Are they faster to converge, and how computationally demanding is each epoch? Reporting execution times and the hardware used (CPU, GPU, or clusters) would provide valuable context.
>
> We **have included** computational cost analysis, comparing the training cost with **increasing number Fourier bases** **(see Figure 9)**, as well as reporting execution times and hardware/software specifics **(see Section 6.2)**. This will provide readers with a clearer understanding of the trade-offs involved.
>
> > The approach becomes computationally infeasible in higher dimensions, as stated in eq 22, which is a significant drawback. Meshless methods are expected to be advantageous in high dimensions, yet this method appears to struggle with scalability.
>
> Thank you for pointing this out. We **have acknowledged** this limitation in the manuscript and have **discussed** potential strategies to address the scalability issue in higher dimensions. Specifically, one direction for overcoming this issue is using tensor decomposition techniques, which effectively reduce the computational complexity of high-dimensional problems. These methods decompose high-dimensional tensors into lower-rank structures, alleviating the computational demands. We **have included a detailed discussion** in the revised manuscript **exploring how applying** these tensor decomposition techniques **could potentially** improve scalability and reduce computational costs in higher-dimensional settings **(see Section 7)**.
>
> > The introduction of L2 regularization in Eq. 23 is problematic. The claim that L2 regularization promotes sparsity is incorrect. Sparsity is typically achieved with L1 regularization.
>
> Thank you for pointing out the distinction between L1 and L2 regularization with regard to promoting sparsity. We agree that L1 regularization is traditionally associated with sparse solutions. However, in our specific case, we are using L2 regularization in conjunction with a pruning strategy, which has been shown to promote sparsity indirectly by driving coefficients toward a threshold, at which point they can be pruned. This approach has been explored in methods such as the SINDy (Sparse Identification of Nonlinear Dynamics) algorithm (Zhang & Schaeffer 2019), where L2 regularization helps to stabilize the coefficients during optimization before they are pruned, yielding sparse solutions. This combination effectively balances between stability and sparsity, as shown in previous work (Wang et al., 2020).
>
> We chose not to use L1 regularization in this case due to practical challenges, including the instability it can introduce during the training process, particularly when optimizing neural networks for complex systems. L2 regularization, combined with a careful pruning process, offers more stable convergence while still leading to sparse solutions after pruning the small coefficients.
>
> We **have revised** the manuscript to clarify this reasoning and ensure that the discussion of sparsity through L2 regularization with pruning is clearly communicated **(see Section 5.2)**.
>
> > In Eq. 24, the focus is on linear operators, but the paper also addresses nonlinear PDEs, such as the Allen-Cahn equation. The treatment of nonlinearity in Eq. 28 seems inadequate, as it overlooks the nonlinear term, effectively solving a different PDE.
>
> We appreciate the reviewer’s comments on the handling of nonlinearity. We would like to clarify that our method does indeed handle nonlinearities. However, we acknowledge that the current description may not fully convey how the nonlinear terms are treated. We **have revised** the relevant sections to explain our approach more clearly. This **includes** a more detailed justification of how the nonlinear components are incorporated into the solution method **(see Section 5.2)**.

---

> ### Author Response · Authors · 2024-10-23
> **Response to Reviewer HXi7 (Part 4)**
>
> **Regarding the Weaknesses of Both Methods:**
>
> We acknowledge the broader set of concerns raised regarding the comparison with traditional methods (FD, FEM, spectral) and the exploration of alternative PINN approaches. In the revised manuscript, we **have addressed** each of these points as follows:
>
> 1. We **have included** comparisons with traditional method to improve upon our evaluations and **have clarified the statements about computational efficiency (see Figure 2 and Table 1).
>
> 2. We would like to clarify that we are not using the original Random Fourier Features as introduced in Rahimi and Recht (2007-2008). Instead, we are using the implementation based on the method described in the paper “On the Eigenvector Bias of Fourier Feature Networks: From Regression to Solving Multi-Scale PDEs with Physics-Informed Neural Networks,” which introduces a specific Fourier feature embedding to address multi-scale problems more effectively. However, we do acknowledge that the figures might have been confusing, and **have revised** the figures to be more interpretable. **Specifically, we have updated the box plot figures in the main text to only include the models achieving the best results, and leave the full box plots in the Appendix.**  We hope this clarification resolves the confusion, and **have ensured** the revised version of the manuscript properly explains the version of RFF-PINNs we compare against.
>
> 3. We **have expanded** the discussion to include recent advancements in random feature and randomized neural network methods, such as RandONet, which have demonstrated improved efficiency for certain PDE problems **(see Section 6.1). Specifically, we do not compare against the methods listed as the proposed Fourier PINN is a modular and general framework that could, in principle, be integrated with or adapted to these alternative methods. For example, the inclusion of Fourier bases within random feature methods, such as RVFLNs, or randomized operator learning frameworks like RandONet, could potentially enhance their ability to capture high-frequency components of the solution. However, investigating such combinations lies beyond the scope of the current work and represents an exciting avenue for future research.**
>
> 4. We **have provided** additional details on the spectral method used and clarified the implementation **(see Section 6.1)**.
>
> 5. We **have investigated** the impact of varying network architectures and report the detailed convergence behavior for different initializations.
>
> 6. We **have clarified** how the relative $\ell_2$ error is computed **in Section 3 paragraph 3, and clarify that all errors reported are on separate testing datasets not seem during training.**
>
> Finally, we **have corrected** the inconsistencies in the domain definitions while clarifying some of our assumptions. We **have** also provided a more detailed description of the experimental design and hyperparameter settings used during the training process to improve reproducibility. **In particular, we have included tables for each experiment in the Appendix listing all hyperparameter and training details.**

---

### Review · Reviewer_mVk8 · 2024-09-29

**Summary Of Contributions:**

The authors explore using physics informed neural networks (PINNs) for PDEs and how they handle balancing accuracy at the boundary vs accuracy inside the domain. The authors present and analyze a new method of Strong Boundary Condition PINNs that do a better job in dealing with higher frequency target problems. The authors then create a new model, the Fourier-PINN, to handle PDEs with high frequencies and benchmark it against a variety of state-of-the-art methods.

**Audience:**

Yes

**Broader Impact Concerns:**

- None

**Claims And Evidence:**

Yes

**Requested Changes:**

Major
- How does $\phi_{\text{exp}}(x)$ work when not in the 1-D setting?
- Figure 2: What value of $\alpha$ do you use for $\phi_{\text{exp}}$? How do you pick your value of $\alpha$?
- What is the goal of the $L_2$ regularization term in your loss function? It is true that $L_2$ regularization will penalize large coefficients. However, this will not promote sparsity, in fact it is likely to do just the opposite. If you want sparsity, you probably want to be using $L_1$ regularization. If you want sparsity in the Fourier basis coefficients, why are you also applying the regularization to the coefficients of the last layer of the Neural Network? Is this just required because otherwise the joint optimization will favor the neural network over the Fourier bases?
- Some of the choices of the optimization algorithm seem strange to me. The weirdest part in my mind is why the authors are alternating between optimizing all parameters with ADAM, and then only optimizing the last layer coefficients with least squares and pruning the bases. The losses for both steps seem very similar, and typically a single joint optimization strategy will outperform an alternating one. So I am curious how Algorithm 1 compares to just running ADAM over all parameters (possibly with the $L_1$ regularization instead of $L_2$).
- How does the performance of the method compare with bases pruning vs without?

Minor
- Top of page 3, "frequencies uniformly sampled from an extensive pre-set range" (also middle of page 9). This makes me think they are sampled randomly, but they are actually all the frequencies $1,\ldots,K$.
- Top of 10, "we apply the cross-product" I have not heard cross-product used in this way, I think it would be better to refer to it as the "outer product" or "tensor product" (as you do later in the sentence) to avoid confusion with the physics cross-product.
- In equation (24), I think it would be more clear to use the notation your paper, rather than the notation of Cyr et al. (2020), i.e. use $w$ for the final layer coefficients.
- Minor typos on page 11, "essense" -> "essence" and "continues" -> "continuous"
- One idea for a future work: You could also allow the frequencies to be optimized, which may allow you to start with fewer.

**Strengths And Weaknesses:**

Strengths
- The new adaptive distance function $\phi_{\text{exp}}$ for boundary condition PINNs is well motivated and well analyzed
- The exposition for the Fourier-PINN is clear.
- The comparison of the Fourier-PINN to other state-of-the-art methods is thorough and convincing for the class of PDEs that the authors care about (high-frequence and multi-frequency).

Weaknesses
- I found some of the notation hard to follow in the Fourier analysis section. Perhaps if equation (15) was in the integral form, or there was an additional equation in the integral form that you substituted the various distance functions into.
- In the Fourier PINN, the Fourier bases grow exponentially in the problem dimension. The authors fully discuss this limitation and leave it to future work.
- The ground truth equations for the numerical experiments consist only of sums and products of sines and cosines. However, the authors state that high-frequency and multi-frequency solutions are the focus of their method.

---

> ### Author Response · Authors · 2024-10-23
> **Response to Reviewer mvK8 (Part 1)**
>
> We thank the reviewer for their positive comments on the adaptive distance function, the exposition of the Fourier-PINN, and the thoroughness of our comparisons to state-of-the-art methods. We appreciate the feedback, which will help us further strengthen the manuscript.
>
> **Regarding the Weaknesses:**
>
> > I found some of the notation hard to follow in the Fourier analysis section. Perhaps if equation (15) was in the integral form, or there was an additional equation in the integral form that you substituted the various distance functions into.
>
> Thank you for your suggestion. We understand that the notation in the Fourier analysis section could be more intuitive. **We maintain the discrete form as it aligns directly with the numerical computations, where the solution domain is discretized. Representing the distance functions in discrete form ensures consistency between the analysis and the numerical implementation. However, we have added a footnote describing the connection between the discrete and integral form on page 11.**
>
> > In the Fourier PINN, the Fourier bases grow exponentially in the problem dimension. The authors fully discuss this limitation and leave it to future work.
>
> We acknowledge this limitation. We **have expanded** the discussion in the conclusion to outline more concrete future directions for addressing the curse of dimensionality, such as the use of sparse Fourier representations or alternative basis functions that scale more efficiently in higher dimensions **(see Section 7)**.
>
> > The ground truth equations for the numerical experiments consist only of sums and products of sines and cosines. However, the authors state that high-frequency and multi-frequency solutions are the focus of their method.
>
> We agree that our current results overly emphasize trigonometric functions. In the revised version, we **have added** numerical experiments using more diverse, non-periodic functions to demonstrate the broader applicability of our approach to high-frequency and multi-frequency solutions beyond just sines and cosines. **Specifically, we have added an experiment on the 1D non-steady state Allen Cahn equation and have included additional analysis problems on examples that aren't fully trigonometric (see Sections 6.6 and 4.1)**

---

> ### Author Response · Authors · 2024-10-23
> **Response to Reviewer mVk8 (Part 2)**
>
> **Regarding the Requested Changes:**
>
> >  How does \( \phi_{\text{exp}}(x) \) work when not in the 1-D setting?
>
> We **have included** a discussion in the revised manuscript on how this adaptive distance function can be generalized to multi-dimensional settings **(see Equation 11)**.
>
> > Figure 2: What value of \( \alpha \) do you use for \( \phi_{\text{exp}} \)? How do you pick your value of \( \alpha \)?
>
> The value of \( \alpha \) is optimized with all other neural network parameters during training, however, the initialization is a hyper-parameter chosen at the start of training. We **have further explained** the choice of initialization in the revised manuscript **(see Section 3 paragraph 3, Section 2.2 paragraph 5)**.
>
> > What is the goal of the L2 regularization term in your loss function? It is true that L2 regularization will penalize large coefficients. However, this will not promote sparsity, in fact it is likely to do just the opposite. If you want sparsity, you probably want to be using L1 regularization. If you want sparsity in the Fourier basis coefficients, why are you also applying the regularization to the coefficients of the last layer of the Neural Network? Is this just required because otherwise the joint optimization will favor the neural network over the Fourier bases?
>
> Thank you for pointing this out. Please see our response to reviewer Hxi7 addressing this concern. We chose to regularize the coefficients in the last layer of the NN as well to penalize the large coefficients in the NN which would potentially prevent the Fourier layer from capturing the high-frequency components in the solution. Additionally, the L2 regularization and truncation leads to a more stable training process. We **have expanded** upon these specific design choices in the revised manuscript to adequately motivate our training routines and architecture design **(see Section 5.2)**.
>
> > Some of the choices of the optimization algorithm seem strange to me. The weirdest part in my mind is why the authors are alternating between optimizing all parameters with ADAM, and then only optimizing the last layer coefficients with least squares and pruning the bases. The losses for both steps seem very similar, and typically a single joint optimization strategy will outperform an alternating one. So I am curious how Algorithm 1 compares to just running ADAM over all parameters (possibly with the L1 regularization instead of L2).
>
> We appreciate the reviewer's suggestion. The alternating optimization was originally chosen to decouple the optimization of the neural network and the Fourier coefficients, allowing for better convergence in certain cases. However, we agree that a joint optimization strategy could offer a simpler and potentially more effective solution. We **have included** a discussion on this comparison in the revised manuscript **along with an additional ablation study comparing the performance of Fourier PINNs using the proposed alternating optimization strategy to joint optimization over all parameters using Adam (see Figure 9). This experiment shows that the alternating routine is crucial for model performance.**
>
> > How does the performance of the method compare with and without basis pruning?
>
> We **have addressed** this in the revised version by conducting additional experiments to compare the performance of the Fourier-PINN with and without basis pruning **(see Figure 13).** **This experiment shows the benefit of the basis pruning and coefficient regularization.**

---

> > ### Comment · Reviewer_mVk8 · 2024-11-04
> >
> > Thank you for your responses. The explanation of L2 + pruning as a method for enforcing sparsity was very helpful in particular.

---

> ### Author Response · Authors · 2024-10-23
> **Response to Reviewer mVk8 (Part 3)**
>
> **Regarding the Minor Changes**
>
> 1. Thank you for pointing this out. We **have updated** the text to make it clear that all frequencies from 1 to \( K \) are used, rather than being randomly sampled, and ensured consistency throughout the manuscript.
>
> 2. We **have revised** the manuscript to use to avoid confusion and maintain consistency in terminology.
>
> 3. We **have revised the description and notation of the adaptive basis selection algorithm for clarity. Specifically, we retain the use of $\textbf{c}$ in the description of the least-squares problem solved in Cyr et al, (2020) (Equation 27 in the revised manuscript---formally Equation 24) as this is referring to the neural network basis coefficients, while Equation 28 (formally Equation 25) uses $\textbf{w}$ as these are all basis coefficients in the Fourier PINN model (which includes both the neural network basis coefficients $\textbf{c}$ along with the Fourier basis coefficients)**.
>
> 4. Thank you for catching these typos. We **have corrected** them in the revised manuscript.
>
> 5. We appreciate this suggestion and agree that optimizing the frequencies could lead to improvements in the efficiency and flexibility of the method if successful. We **have included** this in the discussion for future work.
>
>
> We appreciate the thorough review and valuable feedback provided. These comments will significantly enhance the clarity and quality of the manuscript. Thank you for your time and effort.

---

### Review · Reviewer_WjzW · 2024-10-08

**Summary Of Contributions:**

This manuscript introduces some enhancements to existing Physics-Informed Neural Network (PINN) approaches for solving partial differential equations (PDEs), particularly focusing on learning high-frequency components. Standard PINNs struggle with these due to "spectrum bias," where the neural network efficiently learns low-frequency information but fails to capture high-frequency components. The authors first examine strong boundary condition (BC) PINNs, where they introduce a flexible distance function to satisfy the boundary conditions, offering improvement in accuracy compared to the standard approach. By performing a Fourier analysis on the results, the authors conclude that such strong boundary conditions help with learning higher frequencies, but only to a certain amount, leading to the motivation and introduction of the Fourier PINNs. Fourier PINNs augment standard PINNs with pre-specified, dense Fourier bases, significantly improving their ability to capture high-frequency components across different boundary conditions and problem domains. They also include an adaptive basis selection algorithm during training, that automatically prunes the basis functions/frequencies that are not relevant to the problem. The authors demonstrate this with a series of numerical experiments with different 1D and 2D PDEs, and compare to several competing PINN and classical approaches. The results show that the Fourier-PINNs consistently outperform other methods, especially when faced with problems where high frequencies are relevant.

**Audience:**

Yes

**Broader Impact Concerns:**

No broader impact concerns to report.

**Claims And Evidence:**

No

**Requested Changes:**

Requested changes:
- Include some additional PDE problems, particularly in the 2d case. The authors should choose one or two benchmark problems that do not have an inherently harmonic structure but still contain high frequencies. (very important)
- Include more detailed explanations on what each of the example PDE problems is aiming to evaluate, and a discussion specific to each of the problems selected. My suggestion would be to separate each of the scenarios into its own subsection, and discuss the purpose of the specific numerical problem as well as present and discuss the results in the same section. This can be finally followed by a general discussion and conclusion as the authors have done. (very important)
- Include benchmarks for the introduced BC-PINN with adaptable distance function as introduced in section 2. (important)
- Another interesting variant to evaluate would be the Fourier-PINN with no regularization or basis pruning. This would clearly demonstrate the effects of the basis selection method introduced by the authors (important)
- Fix references to include proper capitalization, and refer to the actual publication venue of Arxiv papers which have already been published after a peer review. (important)
- Ideally, the code should be included for review as well, at least for the BC and Fourier-PINN, with some examples to reproduce the results in the paper. This is crucial to foster transparency and reproducibility in our community (very important).

**Strengths And Weaknesses:**

Strengths:
- The paper provides a strong background and an accessible introduction to PINNs, useful for readers outside of the field.
- The Fourier analysis conducted for the BC-PINNs is well-executed and offers a solid theoretical foundation for the work. This analysis motivates the inclusion of Fourier bases to address the "spectrum bias" problem effectively.
- The concept of Fourier PINNs is both simple and highly effective, demonstrating significant and stable performance across a variety of benchmark PDE problems. The adaptive selection mechanism for the Fourier bases is a practical innovation  enhances performance.


Weaknesses:
- The primary limitation is in the evaluation methodology. The authors test the Fourier PINNs on a limited set of problems (three one-dimensional and two two-dimensional problems). All these problems seem tailored to the strengths of Fourier PINNs, as their solutions largely consist of harmonic functions. While these functions are typical for benchmarks, and gives the authors the ability to control the prevalence of high frequencies, this choice may give a somewhat one-sided view of the method's performance.
- The evaluation lacks sufficient explanation regarding the purpose of the chosen numerical problems. The authors briefly explain this at the beginning of the experiment section, but this is not detailed enough to give clear motivation for the different evaluations. The authors should elaborate on what aspects of network performance are being tested with each problem and how the problems are designed to introduce high frequencies. This would add depth to the evaluation section.
- There is no comparison with the previously introduced BC-PINN with adaptable distance functions, which could have demonstrated the benefits of including Fourier components more clearly.
- The discussion is also underdeveloped. The authors provide a large number of figures, but the accompanying analysis is insufficient, particularly with regard to the different families of equations used in the experiments.
- The reference section is poorly formatted, with inconsistent capitalization and incorrect references for papers already published in journals but listed with Arxiv citations.

---

> ### Author Response · Authors · 2024-10-23
> **Reply to Reviewer WjwzW (Part 1)**
>
> We appreciate the positive feedback regarding the background and introduction to PINNs, along with the theoretical foundation provided by the Fourier analysis and the adaptive basis selection.
>
> **Regarding the Weaknesses:**
>
> > The primary limitation is in the evaluation methodology. The authors test the Fourier PINNs on a limited set of problems (three one-dimensional and two two-dimensional problems). All these problems seem tailored to the strengths of Fourier PINNs, as their solutions largely consist of harmonic functions. While these functions are typical for benchmarks, and gives the authors the ability to control the prevalence of high frequencies, this choice may give a somewhat one-sided view of the method's performance.
>
> We agree that expanding the evaluation to include more diverse PDE problems, particularly those that do not inherently have harmonic structures, would provide a more comprehensive view of the method’s capabilities. We **have added** experiments on additional benchmark problems, including non-harmonic PDEs such as the non-steady-state Allen-Cahn equation. This **helps** showcase how Fourier PINNs perform on problems with high-frequency components that are not purely harmonic in nature **(see Section 6.6)**.
>
>
> > The evaluation lacks sufficient explanation regarding the purpose of the chosen numerical problems. The authors briefly explain this at the beginning of the experiment section, but this is not detailed enough to give clear motivation for the different evaluations. The authors should elaborate on what aspects of network performance are being tested with each problem and how the problems are designed to introduce high frequencies. This would add depth to the evaluation section.
>
> Thank you for pointing this out. In the revised manuscript, we **have** restructure the experimental section to include more detailed explanations of the purpose of each numerical problem. Each scenario **is** presented in its own subsection with a specific discussion on the aspects of network performance being evaluated, the nature of the high-frequency components introduced, and how these problems test the method’s capabilities. This provides clearer motivation for the choices and help readers understand the broader applicability of the method.
>
>
> > There is no comparison with the previously introduced BC-PINN with adaptable distance functions, which could have demonstrated the benefits of including Fourier components more clearly.
>
> We appreciate this suggestion and **have included** comparative benchmarks for the BC-PINN with adaptable distance functions in the revised manuscript. This comparison **highlights** the benefits of including Fourier components, particularly in problems with high-frequency components.
>
> > The discussion is also underdeveloped. The authors provide a large number of figures, but the accompanying analysis is insufficient, particularly with regard to the different families of equations used in the experiments.
>
> We agree that the discussion can be expanded better to interpret the results across the different families of equations. In the revised manuscript, we **have enhanced** the discussion to provide a more detailed analysis of the performance across different problem families, including a breakdown of why certain problems highlight specific strengths or limitations of Fourier-PINNs. We **have also included** more thorough discussions following each result section.
>
> > The reference section is poorly formatted, with inconsistent capitalization and incorrect references for papers already published in journals but listed with Arxiv citations.
>
> We **have corrected** all formatting issues in the reference section. Thank you for pointing this out.

---

> ### Author Response · Authors · 2024-10-23
> **Reply to Reviewer WjwzW (Part 2)**
>
> **Regarding the Requested Changes**
>
> > Include some additional PDE problems, particularly in the 2d case. The authors should choose one or two benchmark problems that do not have an inherently harmonic structure but still contain high frequencies. (very important)
>
> We agree with the importance of this suggestion. As mentioned earlier, we **have included** additional results for problems that do not have inherent harmonic structures. **Specifically, we have included** results from the Allen-Cahn equation, which involves nonlinear dynamics and does not have an inherent harmonic structure **(see Section 6.6)**.
>
> > Include more detailed explanations on what each of the example PDE problems is aiming to evaluate, and a discussion specific to each of the problems selected. My suggestion would be to separate each of the scenarios into its own subsection, and discuss the purpose of the specific numerical problem as well as present and discuss the results in the same section. This can be finally followed by a general discussion and conclusion as the authors have done. (very important)
>
> Thank you for the suggestion. We **have separated** each example into its own subsection and provided a more detailed explanation of the role in evaluating network performance. Each subsection **discusses** the specific challenges it introduces and the corresponding results. We agree that this structure **greatly improves** the clarity of the experimental section and give readers a better understanding of the significance of each experiment.
>
> > Include benchmarks for the introduced BC-PINN with adaptable distance function as introduced in section 2. (important)
>
> As noted earlier, we **have added** a comparison between BC-PINN with adaptable distance functions and Fourier-PINN.
>
> > Another interesting variant to evaluate would be the Fourier-PINN with no regularization or basis pruning. This would clearly demonstrate the effects of the basis selection method introduced by the authors (important)
>
> Thank you for the suggestion. We **have included** an additional ablation experiment where the Fourier-PINN is evaluated without basis pruning or regularization **(see Section 6.5)**. We agree that this **demonstrates** the specific impact of the adaptive basis selection and highlight the improvements it brings regarding performance.
>
> > Fix references to include proper capitalization, and refer to the actual publication venue of Arxiv papers which have already been published after a peer review. (important)
>
> We **have updated** the reference section to ensure the correct formatting.
>
> > Ideally, the code should be included for review as well, at least for the BC and Fourier-PINN, with some examples to reproduce the results in the paper. This is crucial to foster transparency and reproducibility in our community (very important).
>
> We fully agree with the importance of transparency and reproducibility in our community. We will make the code available, including examples for both BC-PINN and Fourier-PINN, to allow for the reproduction of the results presented in the manuscript.
>
>
> We greatly appreciate the reviewer’s detailed feedback and valuable suggestions. The requested changes significantly enhance the clarity, depth, and impact of the manuscript, and we believe these improvements will strengthen the overall contribution of our work. Thank you for your time and effort.

---

### Decision · Action_Editor_VUuN · 2024-11-22

**Recommendation:** Accept with minor revision

**Comment:**

This manuscript received the careful evaluation of three reviewers. My summary of their feedback is the following:

**Strengths**
* The problem studied here is important and relevant to the PINs literature, and the presentation is clear and instructive.
* Their proposal and solutions are well-motivated and implemented.
* The advantage of Fourier PINs is evident.

**Weaknesses**

The contribution is somewhat limited, in the sense that:
* The authors only study Dirichlet BCs, but the method is not applicable to more complex alternatives like Neumann or non-linear boundary conditions.
* One of the advantages of learned approximations to solving PDE is the better scaling with the problem dimension. Yet, the approach presented here scales poorly with this parameter and is thus limited to small dimensional problems.
* The empirical evaluation is limited to 1D and 2D simple problems, mostly with strong harmonic components, and it is thus unclear whether the method's benefits will carry as significantly to more general settings.

### Conclusion

The responses from the authors have been satisfactory, in the sense that they acknowledge the limitations of the proposed method and mention that these will be made very clear in the revised manuscript. Furthermore, the authors will also carry out further experiments in other systems without such explicit components of harmonics, in addition to commenting on potential extensions to address the issues of scaling with dimensions and more general boundary conditions (as well as the symmetric constraint on the steepness parameter).

There is some discrepancy between the recommendations of the three reviewers. However, the consensus is that if the authors implemented all of the changes that they mentioned in their responses, the majority of them would be happy to recommend this paper for acceptance alluding to the strengths highlighted above. Therefore, I am recommending acceptance conditioned on these changes being made. While some of these changes are non-negligible (especially the experimental ones), I regard these as "minor" in the sense that the overall methodology and motivation will remain the same, but I expect the authors to consider serious modifications to address the reviewers' questions and concerns.

**Audience:**

This contribution will be of interest to a significant part of the TMLR community, particularly those in the parametric approximation of solvers for PDEs.

**Claims And Evidence:**

This paper studies the limitations of Physics Informed Networks (PINs) in dealing with boundary conditions. The paper argues that imposing boundary conditions (BC) through distance functions is insufficient since PINs have difficulties learning the high-frequency components needed to better account for BCs. Thus, they propose a Fourier alternative by combining PINs with a Fourier basis, with learnable coefficients. The approach effectively improves fit, and comparables favourably with alternative methods.

All claims are carefully argued for or supported by empirical evidence.